# COOM: A Game Benchmark for Continual Reinforcement Learning

**Tristan Tomilin[1]   Meng Fang[2,1]   Yudi Zhang[1]   Mykola Pechenizkiy[1]**

[1]Eindhoven University of Technology   [2]University of Liverpool

{t.tomilin,y.zhang5,m.pechenizkiy}@tue.nl

Meng.Fang@liverpool.ac.uk

## Abstract

The advancement of continual reinforcement learning (RL) has been facing various obstacles, including standardized metrics and evaluation protocols, demanding computational requirements, and a lack of widely accepted standard benchmarks. In response to these challenges, we present COOM (**C**ontinual D**OOM**), a continual RL benchmark tailored for embodied pixel-based RL. COOM presents a meticulously crafted suite of task sequences set within visually distinct 3D environments, serving as a robust evaluation framework to assess crucial aspects of continual RL, such as catastrophic forgetting, knowledge transfer, and sample-efficient learning. Following an in-depth empirical evaluation of popular continual learning (CL) methods, we pinpoint their limitations, provide valuable insight into the benchmark and highlight unique algorithmic challenges. This makes our work the first to benchmark image-based CRL in 3D environments with embodied perception. The primary objective of the COOM benchmark is to offer the research community a valuable and cost-effective challenge. It seeks to deepen our comprehension of the capabilities and limitations of current and forthcoming CL methods in an RL setting. The code and environments are open-sourced and accessible on GitHub.

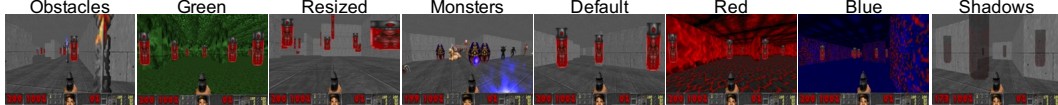

(a) **CD8 Sequence**. The agent is expected to locate the correct targets in the environment and eliminate them. The tasks are based on a single scenario (`Run and Gun`), but the textures and in-game entities vary. The agent needs to effectively adapt its policy to adjust to the visual distortions.

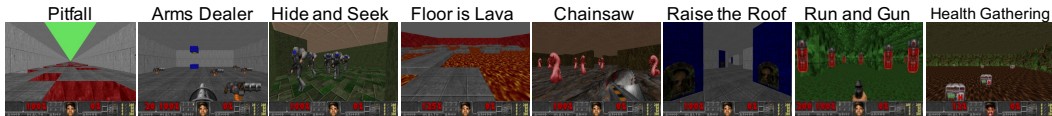

(b) **CO8 Sequence**. In addition to visual disparities between scenarios, the agent is presented with novel diverse goals when exposed to a new task. These objectives are often contrasting, e.g., whereas one task requires seeking enemies, another favours avoiding them entirely.

Figure 1: **COOM sequences** (encompassing tasks ordered from left to right) for task-incremental learning are composed of diverse environments built in ViZDoom [25]. The agent perceives its surroundings from an embodied perspective. The 4-task sequences (**CD4** and **CO4**) are constituted by environments in the second half, whereas **CD16** and **CO16** repeat the original sequence.

37th Conference on Neural Information Processing Systems (NeurIPS 2023) Track on Datasets and Benchmarks.

# 1 Introduction

Deep reinforcement learning (RL) has made immense leaps forward over the past decade in the domain of video games [68, 60, 9], robotic manipulation [28, 30, 72, 50, 76], embodied AI [65, 12, 63], and foremost on a variety of platforms intended for RL research [44, 22, 5, 63]. However, this success has primarily resulted from fine-tuning agents to solve specific tasks. As RL is increasingly applied to solving real-life problems in industry, healthcare, or robotics, situations arise where the environment and conditions are subject to rapid change. Whereas humans are able to learn to perform new similar tasks seamlessly, this competence is still predominantly absent in RL agents, who tend to exceedingly overfit to new tasks and forget all previously acquired competencies. A central goal of RL is thus to build systems that can master a spectrum of skills in environments as noisy and diverse as the real world, while being capable of continuing to learn.

Continual learning (CL) is the ability to swiftly learn consecutive tasks while maintaining adequate performance on previously mastered problems, where they ought to reemerge [49]. The CL paradigm emerges in RL when an agent proceeds to interact with an altered environment in the attempt to refine its policy [54]. Whereas task boundaries in CL might be gradual and smooth, we propose to consider learning a sequence of tasks with sudden transitions. Evaluating a CL agent usually involves several criteria: catastrophic forgetting, forward transfer and backward transfer [59]. CL research is particularly vital as it endorses the AI to thrive in encountered scenarios and conditions, which is much desired for AGI, compared to just solving individual problems.

Many prominent CL platforms are predominantly meant for supervised learning and require non-trivial adaptation for RL [47, 38, 16]. Video games act as ready-made simulation environments, which is why RL research platforms and benchmarks have widely been based on repurposed game engines [7, 6, 19]. There are, however, relatively few visual-based environments for CRL compared to regular RL. Researchers must instead use ad-hoc adaptations of RL benchmarks to achieve the desired setting. Proper virtual 3D simulation platforms and benchmarks are required to facilitate image-based CRL research. They need to be 1) lightweight, 2) well documented, 3) easy to install and use, 4) equipped with standardized metrics and baseline evaluations, and 5) computationally viable to run on small-scale systems and university-level budgets.

To this end, we present COOM (**C**ontinual D**OOM**), a CRL benchmark for embodied pixel-based learning on sequences of tasks with visually distinct 3D environments, designed to assess average performance, forgetting, and forward transfer. The visual modifications manifest in 1) wall and surface textures, 2) types, shapes, and sizes of in-game entities, and 3) modes to render objects. Compared to existing benchmarks for pixel-based learning oriented towards CL, COOM additionally includes a diverse set of objectives. To the best of our knowledge, this is the first benchmark specifically targeted towards CRL in complex 3D environments with differing objectives and visuals.

The contributions of our work are three-fold: 1) We assemble 6 task sequences of different lengths, half with task-dependent objectives, an aspect often absent in previous works. We further include a very complex sequence, intended as a challenge to the RL community to be solved with more advanced future algorithms. We include a demo of a trained agent performing COOM tasks. 2) We design 8 novel ViZDoom scenarios of two difficulties with contrasting visuals and dynamics to compose and publicly release COOM, a Continual RL benchmark. 3) Following the CL evaluation principles from [70], we employ multiple well-known CL methods for baseline evaluations on our task sequences, assessing prominent CL criteria. This makes our work the first to benchmark image-based CRL in 3D environments. We demonstrate how several methods fail to achieve the CL desiderata and how none come close to solving the complex sequence. We further elaborate on how the action distribution provides a more comprehensive understanding of the agent's continual progression. We conclude that critic regularization and single output head architectures tend to diminish performance in CL.

# 2 Related work

Most recent eminent CL platforms and frameworks, such as Sequoia [47], Avalanche [38], and Continuum [16] learn from static data sets of fixed sizes [35], and are thus not predominantly intended for RL. In CRL, the data sequence consists of different environments, and data samples are obtained through the agent-environment interaction. **Sequoia** introduces metrics and baselines aimed at CRL but only consists of simple environments like state-based manipulation tasks from

Table 1: Comparison of existing Continual Reinforcement Learning benchmarks with COOM.

| Benchmark | 3D | No. Task Sequences | Embodied View | Vision Transfer | Objective Transfer | Unified Metric |
|---|---|---|---|---|---|---|
| Continual World [70] | ✓ | 2 | ✗ | ✗ | ✓ | ✓ |
| L2Learner [23] | ✓ | 0 | ✓ | ✓ | ✓ | ✗ |
| Jelly Bean World [52] | ✗ | 0 | ✗ | ✓ | ✗ | ✗ |
| CORA [53] | ✓/✗ | 4 | ✓/✗ | ✓/✗ | ✓/✗ | ✓ |
| CRLMaze [37] | ✓ | 4 | ✓ | ✓ | ✗ | ✓ |
| COOM | ✓ | 7 | ✓ | ✓ | ✓ | ✓ |

**Meta-World** [73], continuous control tasks from MuJoCo [66], and Monster Kong [62]. **Avalanche RL** [39] introduces a library for CRL, but only includes direct support for basic Atari [43] and **Habitat v1.0** [58] environments. Moreover, it does not present any experimental results on baseline methods.

Most prominent modern RL platforms and benchmarks do not offer a fixed setting for CL [19, 24, 13, 63, 61, 51, 75, 14, 2]. The **Atari** benchmark [7] has often been employed to assess popular CL methods [27, 59, 55]. However, the games on the platform are deeply unrelated, lacking the potential for transfer. **DeepMind Lab** [6] provides a number of diverse 3D environments and **MineRL** [19] incorporates many of the elements found in lifelong learning [64]. Both have widely been used for assessing CL methods [56, 26, 55, 64]. However, neither include fixed task sequences or standardized CL metrics, requiring users to hand-pick existing tasks or design new environments. This causes a tedious experimental setup and provides no evident means of comparison to other works. **DeepMind Lab** and **ProcGen** [13] use procedural content generation (PCG), which renders them computationally expensive to run for CL. Platforms like AI2Thor [29] and iGibson [36], based on the Unity3D game engine, are good at replicating the complexity and visual fidelity of real-world problems. However, they are computationally expensive to simulate, and thus often infeasible for low-budget computational setups.

Previous CRL benchmarks often exhibit certain deficiencies. **CRLMaze** [37], based on ViZDoom, presents a non-stationary object-picking task, subject to visual changes. However, it lacks a comprehensive baseline evaluation of the most popular CL methods, does not undertake to change the objective, and only modifies three attributes (light, textures, objects). **Continual World** [70] assembles robotic manipulation tasks from Meta-World [73] to evaluate CRL and **Jelly Bean World** [52] uses PCG to generate a 2D gridworld test bed. However, neither addresses embodied AI nor image-based learning. **L2Learner** [23] creates a 3D PCG world to assess embodied agents, but lacks baseline evaluations and only consists of 5 tasks in a single environment. **CORA** [53] aggregates environments from Atari [7], ProcGen [13], MiniHack [57], and CHORES [29] into task sequences. The first three environments are all 2D, and the latter lacks diverse visuals. We refer readers to Table 1 for the comparison. Numerous metrics have been employed throughout CRL research that attempt to measure similar objectives [15, 10, 48]. The abundance and incoherence of metrics can create confusion in interpreting and disentangling results.

## 3 Preliminaries

**Reinforcement Learning from Images**   We formulate embodied image-based learning as a partially observable Markov decision process (POMDP) [8], in which an agent interacts with an environment over a fixed horizon of discrete time steps $T$. A POMDP can be described as a tuple $(\mathcal{S}, \mathcal{A}, p, R, \Omega, O, \gamma)$. At each timestep $t$, the environment is in some state $s_t \in \mathcal{S}$. By taking an action $a_t \in \mathcal{A}$, the agent causes the environment to transition to another state $s_{t+1}$ with probability $p = \mathbb{P}(s_{t+1}|s_t, a_t)$. The agent cannot observe the full state $s_t$ of the environment, but an observation $o_t \in \Omega$ dependent on the action $a_t$ taken and new state $s_{t+1}$ reached with probability $O(o|s_{t+1}, a)$. $\Omega$ is the high-dimensional set of pixel-observations. The agent receives a reward $r_t = R(s_t, a_t)$, which is mapped by the reward function $R : \mathcal{S} \times \mathcal{A} \to \mathbb{R}$ given a state and action. Finally, $\gamma \in [0, 1)$ is the discount factor determining how much immediate rewards are favoured over more distant rewards. The $n$-step return $R_{t:t+n}$ at time step $t$ is defined as the discounted sum of rewards, $R_{t:t+n} = \sum_{i=1}^{n} \gamma^i r_{t+i}$. The value function $V^\pi(s) = \mathbb{E}\big[R_{t:T}|s_t = s, \pi\big]$ is the

Table 2: **COOM scenarios.** The core properties determine 1) how the agent's performance is measured; 2) whether enemies exist; 3) does the agent has a weapon; 4) do items spawn on the ground; 5) number of iterations in an episode; 6) which action can the agent execute apart from navigation; 7) what is randomized in the environment.

| Scenario | Success Metric | Enemies | Weapon | Items | Max Steps | Execute | Stochasticity |
|---|---|---|---|---|---|---|---|
| Pitfall | Distance Covered | ✗ | ✗ | ✗ | 1000 | JUMP | Pitfall tile locations |
| Arms Dealer | Weapons Delivered | ✗ | ✓ | ✓ | 1000 | SPEED | Weapon and delivery locations |
| Hide and Seek | Frames Alive | ✓ | ✗ | ✓ | 2500 | SPEED | Enemy behaviour, item locations |
| Floor is Lava | Frames Alive | ✗ | ✗ | ✗ | 2500 | SPEED | Platform locations |
| Chainsaw | Kill Count | ✓ | ✓ | ✗ | 2500 | ATTACK | Enemy and agent spawn locations |
| Raise the Roof | Frames Alive | ✗ | ✗ | ✗ | 2500 | USE | Agent spawn location |
| Run and Gun | Kill Count | ✓ | ✓ | ✗ | 2500 | ATTACK | Enemy and agent spawn locations |
| Health Gathering | Frames Alive | ✗ | ✗ | ✓ | 2500 | SPEED | Health kit spawn locations |

expected return from state $s$, when actions are selected accorded to a policy $\pi(a|s)$. The action-value function $Q^\pi(s, a) = \mathbb{E}\big[R_{t:T}|s_t = s, a_t = a, \pi\big]$ is the expected return following action $a$ from state $s$. We aim to find a policy $\pi(a_t|s_t)$ that maximizes the cumulative discounted return $\mathbb{E}_\pi\big[\sum_{t=1}^T \gamma^t r_t | a_t \sim \pi(\cdot|s_t), s'_t \sim p(\cdot|s_t, a_t), s_1 \sim p(\cdot)\big]$.

**Continual Learning**   In the CL setting of this paper, we aim to learn a policy $\pi_\theta$ by training an agent sequentially on a fixed task sequence $\mathcal{T} = \{\mathcal{T}_1, \ldots, \mathcal{T}_n\}$. The $i^{th}$ task is trained during the interval $t \in [(i-1) \cdot \Delta, i \cdot \Delta]$, where $\Delta$ is the fixed number of iterations per task, during which the agent can only interact with the given environment. The agent thus obtains trajectories $\tau_{i,1}, \ldots, \tau_{i,\Delta}$ from a single task to learn from, during which the POMDP to solve becomes $(\mathcal{S}_i, \mathcal{A}, p_i, r_i, \gamma)$.

## 4   COOM benchmark

The COOM (**C**ontinual D**OOM**) benchmark is based on ViZDoom [25], a flexible RL research platform for learning from raw visual information, based on the engine of the classical FPS video game Doom. ViZDoom has the benefit of being very lightweight, enabling gameplay up to 7000 FPS on a single GPU. The benchmark is comprised of 8 scenarios built in the *Action Code Script* (ACS) language. Every scenario is orientated towards accomplishing a particular objective, each having one or multiple aspects which make the environment stochastic. In Table 2, we display the core properties of each scenario: 1) the metric for measuring agent performance, 2) are there enemies in the environment 3) is the agent equipped with a weapon, 4) do items spawn on the ground, 5) how many iterations an episode lasts, 6) which is the *execute* action, and 7) what is randomized in the environment. A more detailed description of each scenario can be found in Appendix A. The limitations of COOM are addressed in Appendix L.

### 4.1   Basic Setup

**Observations**   The agent is limited to only observing a portion of its surroundings, having a 90 degree horizontal field of view (FoV), which ranks lower than human vision [17] and modern cameras [74]. The observation space $\mathcal{S}$ is 4 stacked frames of $160 \times 120$ pixels in a 4:3 resolution with 3 channels of 8-bit values in RGB, which represents the partially observed environment from the first person perspective of the embodied agent.

**Action Space**   To facilitate the training of a CL task sequence using a single model, we require a unified action space across environments. We thus limit the full actions from the original Doom game to suit the scenarios of our benchmark. We obtain a multi-discrete action space $\mathcal{A}$ for all environments by finding the Cartesian Product of singular actions $\mathcal{A} = A_1 \times A_2 \times A_3$, where $A_1 = \{\texttt{MOVE\_FORWARD}, \texttt{NO-OP}\}$, $A_2 = \{\texttt{TURN\_LEFT}, \texttt{TURN\_RIGHT}, \texttt{NO-OP}\}$, and $A_3 = \{\texttt{ATTACK/USE/SPEED}, \texttt{NO-OP}\}$. The agent hence needs to choose one of $|\mathcal{A}| = 12$ actions per iteration. Note that the execute operation $A_3$ is scenario-dependent as indicated in Table 2.

**Rewards**   The core component of the reward granted in each scenario is directly tied with the success metric (e.g., performance in *Run and Gun* increases when shooting a target, for which the

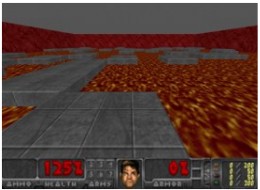 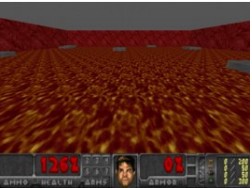 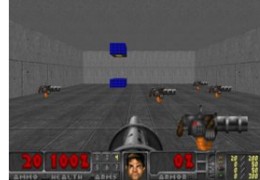 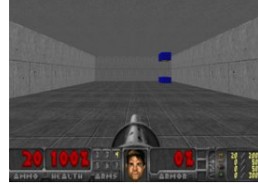

(a) Default environment of *Floor is Lava* (left) and its *hard* counterpart (right), where only 5% of the surface tiles are platforms instead of the regular 30%. The agent is tasked to stand on the platforms, not the lava. The locations of all platforms are randomized after a time interval, which is shorter in the harder version.

(b) Default environment of *Arms Dealer* (left) and its *hard* counterpart (right), where only 1 weapon is spawned at a time instead of the regular 5. The agent is tasked to pick up a weapon and deliver it to the blue platform that only appears after pickup and immediately disappears after delivery.

Figure 2: **Constructing the hard COC sequence**. We modify each CO8 task, leading to more sparse and elusive rewards, to impose a more complex CL challenge.

agent is also rewarded). We further utilize reward shaping for more granular dense feedback. A comprehensive description of reward functions is presented in Appendix B.

## 4.2 Task Sequences

We compose three lengths of sequences: 1) 8 unique tasks $\mathbb{T}^8 = \{\mathcal{T}_0, \ldots, \mathcal{T}_7\}$, 2) including only the second half of the former $\mathbb{T}^4 = \{\mathcal{T}_4, \ldots, \mathcal{T}_7\}$ to streamline the experimental process, considering the computationally intensive nature of training RL agents, and 3) repeating the sequence $\mathbb{T}^{16} = \mathbb{T}^8 + \mathbb{T}^8$ (similar to [70]) allowing for revisiting each task in the same order. By default, the order of tasks in a sequence is predetermined. However, we provide the option for randomizing the sequence order, although this requires a substantially higher number of trials to ensure consistent results. This is due to 1) the performance of most continual learning methods being heavily dependent on the feature representation learned for the first task, and 2) the transferability of knowledge varying among different pairs of tasks. We distinguish between two sequence modalities.

**Cross-Domain Sequence**   In the cross-domain setting, the agent is sequentially trained on modified versions of the same scenario. We select the `Run and Gun` scenario as the basis, as it best resembles the original *Doom* game, requiring the agent to navigate the map and eliminate enemies by firing a weapon. As depicted in Figure 1a, the objective and map layout remain the same across tasks, whereas we modify the environment by 1) changing wall, ceiling and floor textures, 2) varying enemy size, shape and type, 3) randomizing the view height of the agent, and 4) adding obstructions which block the agent's movement. The modifications are inspired by the LevDoom generalization benchmark [67]. We henceforth refer to the cross-domain sequences as CD4, CD8 and CD16.

**Cross-Objective Sequence**   In addition to changing the visuals and dynamics within a scenario, we now design and employ new scenarios with novel objectives for consecutive tasks, as illustrated in Figure 1b. This presents a more diverse challenge, as the goal might drastically change from locating and eliminating enemies (`Run and Gun` and `Chainsaw`) to running away and hiding from them (`Hide and Seek`). This introduces a concept in CL commonly known as *Task Interference*. Similarly, the scenario `Floor is Lava` requires the agent to stay at a bounded location for good performance, whereas scenarios `Pitfall`, `Arms Dealer`, `Raise the Roof`, and `Health Gathering` encourage constant movement. We equivalently refer to the sequences as CO4, CO8 and CO16.

**Cross-Objective Challenge**   There are two main dogmas in the RL community that dictate how to make progress in solving real-world decision-making problems [42]. The first is usually referred to as the *generalizable agent* view, in which focus should be attended towards the large-scale training of agents that solve diverse problems, hoping that along the way, a generalist agent will evolve. Our previously defined two task sequences endorse this philosophy. The second view, generally referred to as *deployable RL*, takes a more pragmatic view by seeking to design RL algorithms that solve particular difficult problems. To facilitate the advancement of the latter principle, we include a task sequence in our benchmark which serves as a challenge, aimed foremost to initially be solved. We posit that harder tasks are easier to forget in a CL setting. To this end, we create a *hard* counterpart

Table 3: Results of Average Performance (**AP**), Forgetting (**F**) and Forward Transfer (**FT**) across 10 seeds. Extended results with 95% confidence intervals are presented in Tables 19 of Appendix N. The result of the best performing method is highlighted in bold.

| Method | CD4 | | | CO4 | | | CD8 | | | CO8 | | | COC | | | Average | | |
|---|---|---|---|---|---|---|---|---|---|---|---|---|---|---|---|---|---|---|
| | AP | F | FT | AP | F | FT | AP | F | FT | AP | F | FT | AP | F | FT | AP | F | FT |
| PackNet | **0.92** | 0.00 | **0.40** | 0.87 | 0.01 | -0.24 | 0.91 | -0.01 | 0.19 | 0.82 | -0.01 | **0.25** | **0.20** | 0.04 | **0.08** | **0.74** | 0.01 | 0.13 |
| MAS | 0.55 | 0.50 | 0.11 | 0.72 | 0.24 | -0.04 | 0.82 | 0.14 | 0.25 | 0.58 | 0.19 | 0.01 | 0.04 | 0.09 | 0.02 | 0.54 | 0.23 | 0.07 |
| AGEM | 0.35 | 0.80 | 0.10 | 0.42 | 0.80 | **0.03** | 0.30 | 0.86 | 0.17 | 0.30 | 0.84 | 0.23 | 0.02 | 0.11 | 0.02 | 0.28 | 0.68 | 0.11 |
| L2 | 0.71 | 0.01 | -0.28 | 0.80 | 0.00 | -0.60 | 0.87 | -0.03 | 0.07 | 0.71 | **-0.04** | -0.32 | 0.10 | **0.00** | 0.00 | 0.64 | **-0.02** | -0.23 |
| EWC | 0.69 | 0.00 | -0.41 | 0.65 | 0.05 | -0.77 | 0.76 | **-0.04** | -0.55 | 0.65 | 0.05 | -0.38 | 0.07 | **0.00** | -0.01 | 0.56 | 0.00 | -0.43 |
| VCL | 0.46 | 0.77 | 0.21 | 0.40 | 0.82 | -0.57 | 0.36 | 0.73 | 0.04 | 0.39 | 0.64 | 0.20 | 0.03 | 0.16 | 0.05 | 0.33 | 0.62 | -0.01 |
| Fine-tuning | 0.59 | 0.63 | 0.32 | 0.50 | 0.71 | -0.01 | 0.45 | 0.75 | **0.28** | 0.44 | 0.48 | 0.23 | 0.02 | 0.10 | 0.02 | 0.40 | 0.53 | **0.17** |
| ClonEx-SAC | 0.87 | 0.00 | 0.11 | 0.86 | **-0.03** | -0.26 | **0.92** | -0.03 | 0.13 | **0.89** | 0.00 | **0.27** | 0.13 | 0.01 | 0.03 | 0.73 | -0.01 | 0.06 |
| Perfect Memory | 0.89 | **-0.01** | 0.30 | **0.89** | 0.02 | **0.03** | - | - | - | - | - | - | - | - | - | - | - | - |

for each task in CO8, as illustrated in Figure 2. We will further refer to this sequence as **COC**, which is further elaborated on in Appendix F.

### 4.3 Evaluation Protocol

The agent is sequentially trained on each task $\mathcal{T}_i$ for $\Delta = 200K$ iterations. After every 1000 iterations, we evaluate the policy on each task of the sequence for 10 episodes. 50K most recent trajectories are stored in the replay buffer, which is emptied at the end of each task by default. The Cross-Domain sequences are based on the *Run and Gun* scenario, in which *Kill Count* is the core measure of performance. In the Cross-Objective sequences, however, the agent is expected to achieve different goals across tasks. We are hence unable to directly use a single metric to adequately compare performance across tasks. As it would be tedious and convoluted to compare the results between methods on each task from a Cross-Objective sequence individually, we define a unified metric $success = f_i(score) \in [0, 1]$, where $score$ represents the task metric (see Table 2) and $f_i$ is a function for task $i$ that transforms the $score$ to $success$. The lower bound $f_i(score) = 0$ on $\mathcal{T}_i$ is determined by the performance of a random agent, and the upper bound $f_i(score) = 1$ by **SAC** trained until convergence.

**Average Performance** $P$ is the core evaluation metric in COOM and is measured across the number of total sequence iterations $T = |\mathcal{T}| \cdot \Delta$ averaging the $success$ of all tasks $\mathcal{T}$ at each time step $t$:

$$P = \frac{1}{|\mathcal{T}|} \frac{1}{T} \sum_{\mathcal{T}_i \in \mathcal{T}} \sum_{t=0}^{T} f_i(score_t). \tag{1}$$

**Forgetting** $F$ quantifies the average decrease of $success$ from the end of training each task $i$ to the end of the entire sequence $T$. For consistency, we consider $k$ last iterations of a task:

$$F = \frac{1}{|\mathcal{T}|-1} \sum_{i=0}^{|\mathcal{T}|-1} \frac{1}{m} \sum_{j=0}^{k-1} f_i(score_{(i+1)\cdot\Delta-j}) - \frac{1}{m} \sum_{j=0}^{k-1} f_i(score_{T-j}). \tag{2}$$

Note that we exclude the final task, as no forgetting can have occurred. In the case of negative values for forgetting ($F < 0$) we observe positive backward transfer.

**Forward Transfer** FT is measured by first finding the area under the training curve $AUC_i$ on task $i$ for the CL method and for the SAC baseline $AUC_i^b$ with

$$AUC_i = \frac{1}{\Delta} \int_{i-1\cdot\Delta}^{i\cdot\Delta} f_i(score_t)\mathrm{d}t, \; AUC_i^b = \frac{1}{\Delta} \int_0^{\Delta} f_i^b(score_t)\mathrm{d}t. \tag{3}$$

We can then first find the average normalized area between the curves across all tasks. Note that we do not consider the first task, as there is no knowledge to transfer forward from previous tasks:

$$FT = \frac{1}{|\mathcal{T}|-1} \sum_{i=1}^{|\mathcal{T}|} \frac{AUC_i - AUC_i^b}{1 - AUC_i^b}. \tag{4}$$

## 5 Experiments

### 5.1 Setup

In this section, we assess the CL desiderata of baselines from three different families of popular CL methods on our task sequences. **Regularization-based** methods constrain weight updates in order to maintain knowledge from previous tasks. **L2** [27] adds a simple $L_2$ penalty to achieve this. **MAS** [3] utilizes a weighted penalty on each parameter depending on its importance to the network

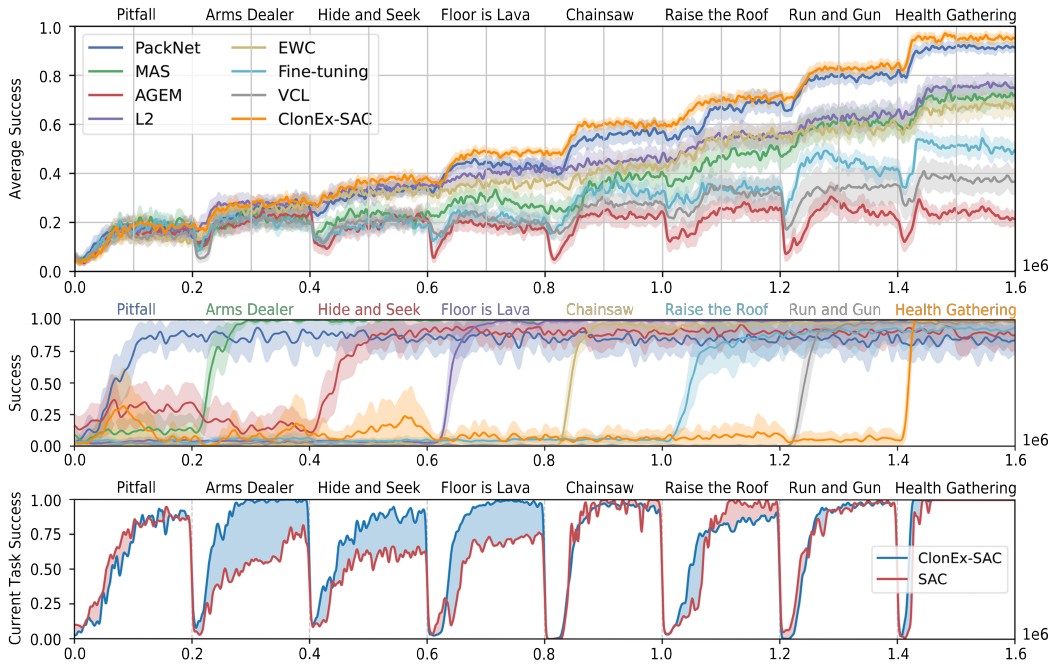

Figure 3: **Visual analysis of CO8 results**. 1) The average cumulative **performance** curves indicate a high disparity across baselines (**top**). 2) ClonEx-SAC effectively mitigates **forgetting** according to the continual evaluation curves of individual tasks (**middle**). The tasks are very diverse, as no notable performance is reached before exposure to the task itself. 3) ClonEx-SAC successfully transfers knowledge forward compared to the vanilla SAC, which is trained on each task from scratch (**bottom**). The blue area between the curves indicates positive **forward transfer** and red represents its negative counterpart.

output. **EWC** [27] uses the Fisher information matrix to approximate the importance of each weight. **VCL** [45] uses variational inference to minimize the KL divergence between the prior and posterior distribution of parameters. **Structure-based** methods preserve and update the network architecture to efficiently learn and adapt to new tasks over time. **PackNet** [41] forbids any changes to parameters that are important for previous tasks by freezing the most relevant weights at the end of each task and pruning the rest. **Rehearsal-based** methods retain samples from previous tasks to constrain forgetting. **Perfect Memory** [70] refrains from resetting the buffer after a task and thus remembers everything. **A-GEM** [11] projects gradients from new samples as to not interfere with previous tasks. **ClonEx-SAC** [71] retains some samples from previous tasks and performs behavior cloning based on them to reduce forgetting. We also include the naïve **Fine-tuning** baseline, which simply continues regular weight updates on a new task. All the CL methods take the efficient and well-known Soft Actor-Critic (**SAC**) [20] as an RL training backbone. We follow the evaluation protocol outlined in Section 4.3 to measure performance of all baselines on CD4, CO4, CD8, CO8 and COC with fixed orders across 10 seeds controlling the pseudorandom nature of the environments. We refrain from using random sequence orders due to the necessity for numerous runs to ensure consistent evaluation, a demand that exceeds our available computational resources. Similarly, we exclude Perfect Memory on 8-task sequences due to the unfeasible memory requirements for storing all trajectories. We grant the agent full access to the task identity both at training and test time, a CL setting coined as *Task-Incremental Learning* [69]. We use $95\%$ confidence intervals for all results. Our framework is discussed in finer detail in Appendix C.

## 5.2 Main Results

**Performance**    PackNet, closely followed by ClonEx-SAC, triumphed across all sequences according to the results in Table 3. We conjecture PackNet's success on our benchmark to two key factors: 1) knowing the task identity during evaluation, and 2) short task sequences with distinct boundaries, as PackNet needs to assign a fixed fraction of parameters to each task. It is noteworthy that the top three best-performing methods (PackNet, ClonEx-SAC, and L2) originate from different families of CL,

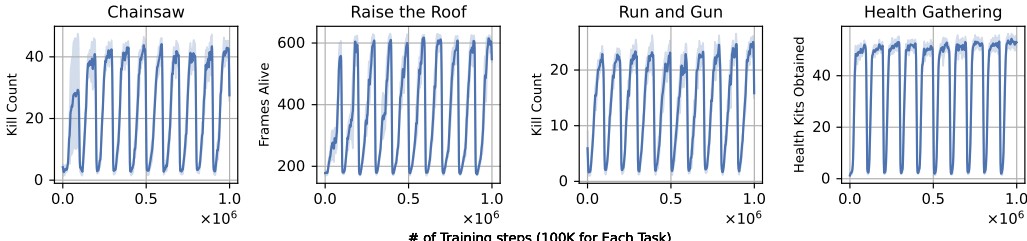

Figure 4: **SAC does not exhibit a loss of network plasticity on COOM.** Measuring the individual performance of each task across 5 seeds displays that repeated exposure to the same tasks throughout 10 iterations of the CO8 sequence does not hinder the capacity to reattain peak performance.

indicating that each approach has a strong potential for achieving excellent results in the benchmark. Although Perfect Memory performed well on the 4-task sequences, it required nearly twice as long to run, and took up 16x more memory in the process. Memory consumption and walltime is further discussed in Appendix E. The average success curves in Figure 3 illustrate a significant disparity in performance among our baselines. Despite an exhaustive exploration of hyperparameters, we were unable to attain satisfactory results with AGEM and VCL. These methods ranked as the poorest performers in our benchmark, remarkably even falling below the naïve fine-tuning approach. Comparing results across sequence lengths, almost all methods showed higher results in CO4 than CO8, whereas among the cross-domain sequences, the top four methods had a substantially higher performance on the longer version. Performance related to sequence length is more extensively analysed in Appendix G. The performance difference between CD and CO sequences are not that staggering, indicating that visual perturbations alone, without a changing objective in our embodied 3D environments, are sufficiently challenging for our CRL baselines.

**Forgetting** The top four baselines excel at preserving performance on previously learned tasks, approaching near-perfection. However, out of these four, L2 and EWC have a substantially lower performance, indicating that regularization-based methods predominantly emphasize stability at the expense of plasticity. Despite an extensive hyperparameter search for weaker regularization, we were not able to achieve a more favorable stability-plasticity trade-off. In particular, a lower regularization constant exhibited a loss of stability but no noticeable further gain in plasticity. In comparison to good stability, AGEM and VCL experience a decline in average performance immediately after losing access to a learned task. In terms of forgetting particular tasks, `Pitfall` and `Arms Dealer` appear the hardest to remember given our fixed ordering of tasks in the CO sequence, as analyzed in Appendix D.

**Transfer** We attribute the core capacity for transfer in our benchmark environments to navigation. Success in most tasks heavily relies on the ability to navigate effectively (e.g., repeatedly running into walls rarely contributes to a high reward). Being able to identify objects and entities from their surroundings also holds significant importance for most tasks (e.g., detecting useful items to collect). The training success curve of ClonEx-SAC in Figure 3 delineates the poor sample-efficiency of the vanilla SAC which is unable to attain comparable performance on several tasks by learning them from scratch. Regularization-based methods fall significantly below others in terms of utilizing acquired competencies to effectively learn future tasks. We did not observe notable backward transfer in our experiments.

## 5.3 Network Plasticity

Recent research has emphasized the importance of network plasticity in RL [1, 40, 46, 33]. To further enhance our analysis and provide a comprehensive perspective on the ability to adapt to new targets on our benchmark. Similar to the experimental design in [1], we aim to investigate whether our base SAC agent experiences a decline in network plasticity when confronted with the Cross-Objective task setting. Intriguingly, unlike the trend observed in [1], our findings diverge given the learning curves of CO4 in Figure 4, where the agent cyclically repeats the sequence 10 times. The agent is able to reach similar results in equal duration throughout the iterations without any noticeable deterioration. We therefore conclude that our base SAC agent does not exhibit a decline in network plasticity when subjected to our benchmark. An avenue for future research would be to investigate whether other

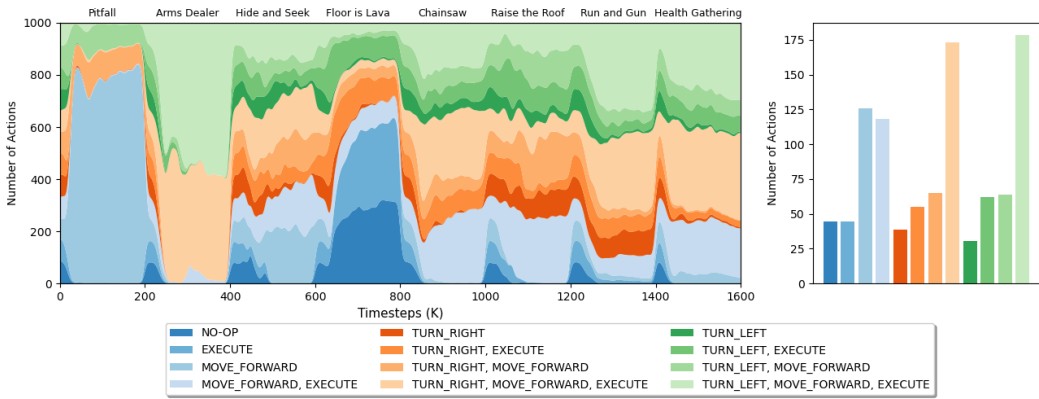

Figure 5: **PackNet quickly adapts its policy to fit a new task when training on CO8**. The stackplot (**left**) depicts the distribution of actions executed in the environment. For example, in `Pitfall`, PackNet highly favours moving forward while accelerating to reap the highest return. The histogram (**right**) indicates the mean number of action executions across the entire sequence. Actions constituting maximal sub-actions appear to be most optimal according to PackNet.

methods, such as model-based approaches, exhibit a similar pattern. The results of CO8 and further analysis is included in Appendix H.

## 5.4 Action Distributions

Since the agent has a fixed multi-discrete action space across all scenarios, we can visualize how the action distribution shifts over time as training progresses to new tasks. We plot the actions executed by PackNet on CO8 in Figure 5. PackNet's high performance suggests it has obtained an effective action distribution. We can observe how PackNet quickly manages to adjust its policy to select appropriate actions when presented with a new task. The action distribution indicates, that on, average executing multiple sub-actions in a single timestep is more beneficial. In Appendix M we further visualize how our baselines are able to maintain the acquired action distribution on each given task.

## 5.5 Method Variations

**Image Augmentation**    Using augmented images as input has been recognized for its potential to enhance data diversity, mitigate catastrophic forgetting, and facilitate adaptation to changes in data probability distribution [21]. We explore the impact of incorporating three established image augmentation methods for RL to our selected baselines on COOM: 1) **Random Convolution** [34] randomly perturbs input observations, 2) **Random Shift** [31] pads each image with 4 pixels (repeating boundary pixels) and then applies a random crop, yielding the original image shifted by $\pm 4$ pixels, 3) **Random Noise** [32] injects Gaussian noise into the image.

In our comparative analysis depicted in Figure 6, we can observe that random shifts and noise have minimal impact on the performance of our baselines, whereas random convolutions significantly degrade all methods except PackNet. They lead to substantial imbalances in the data distribution for initial tasks, causing the policy to converge to a local minimum from which it struggles to recover. This phenomenon is especially pronounced in regularization-based methods that rely on maintaining stable feature representations, which are highly susceptible to disruption caused by random convolutions due to their rigid architecture and stringent regularization. Additionally, these perturbations affect the data distribution stored in the memory of rehearsal-based methods, diminishing their effectiveness for future task learning. Our findings indicate that PackNet can better recover from a degenerate policy stuck in local minima, attributed to freeing up redundant parameters after training on a task. This suggests that architecture-based methods exhibit greater flexibility in the face of significant changes in data distribution.

**LSTM**    We include an LSTM encoder atop the CNN head in our model architecture to assess its impact on our baseline methods. Employing a similar approach to a supervised CL setting has previously been shown to be beneficial [18]. On our benchmark, however, this approach completely

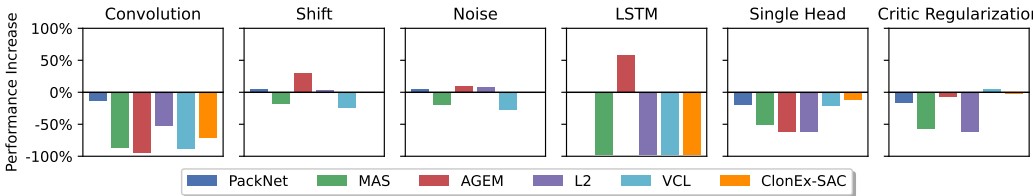

Figure 6: Performance comparison of potential method enhancements. Key observations: 1) While most baselines are unable to learn a single task using an LSTM encoder, AGEM reaps a substantial benefit, and PackNet remains indifferent. 2) Critic regularization has a detrimental effect on CL, particularly on regularization-based methods. 3) Methods performing well on COOM suffer less from only a single head output. 4) Attributed to weight-pruning, PackNet demonstrates superior recovery from a degenerate policy under an imbalanced data distribution with random convolutions.

deteriorates the performance of most methods, rendering them unable to learn anything meaningful. Strikingly, PackNet's performance remained consistent, whereas AGEM experienced a substantial increases. We hypothesize that, by virtue of capturing sequential patterns, an LSTM encoder can facilitate more accurate and efficient retrieval of relevant experiences from the episodic memory. This improved retrieval could enable AGEM to make better use of past knowledge when confronted with new tasks. However, more in-depth analysis is required to confirm this.

**Critic Regularization**    In the context of actor-critic frameworks like SAC, there is a design decision regarding whether to regularize the critic. In some CL methods, the critic is only used during training and only needs to store the value function for the current task, making forgetting less of a concern. However, certain methods, such as replay-based approaches, may not easily accommodate a framework where only the actor is regularized. We thus investigate critic regularization affects our baselines. Figure 6 confirm the findings from [70], that regularizing the critic network has a harmful impact for CL. This suggests that it is better to let the critic adapt freely to the current task while prioritizing the minimizing of forgetting in the actor. However, this means that the critic needs to be trained from scratch, which is not particularly efficient when tasks are repeated.

**Output Heads**    In our default setup, we use a separate output head for each task, which is known to help prevent forgetting [70]. However, this approach is inapplicable to dynamic scenarios with and unknown number of tasks. Figure 6 depicts how a single-head architecture impacts our baselines. All methods show a decline in performance, however ClonEx-SAC, PackNet and VCL manage to better maintain their original performance. Extended results are presented in Appendix I.

## 6    Conclusion

Training proficient agents, who are able to continually adapt to learning new tasks in an altering environment, currently remains one of the greatest challenges in RL. To aid the community in grappling with this challenge, we presented and openly released COOM, a novel benchmark for assessing CL in visually distinct 3D environments with diverse objectives. COOM comprises seven task sequences of varying length, complexity, and modality, effectively demonstrating the strengths and weaknesses of popular continual learning methods. Our experimental evaluations reveal the impact of different variations to baseline methods, highlighting the importance of critic regularization and task-dependent output architectures. We visually illustrate how high-performing methods quickly adapt their policies and maintain learned action distributions across tasks within a sequence. All the methods in our evaluations struggle with the complex COC sequence, making it a valuable testing ground for future research, showcasing both learning challenges and transfer difficulties. COOM serves as a crucial tool to inspire the development of more capable algorithms and we expect it to facilitate the evaluation and comprehension of future continual reinforcement learning methods.

## Acknowledgements

This work was conducted with the assistance of the Dutch national e-infrastructure, generously supported by the SURF Cooperative through grant number EINF-5755.

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

# A  Scenario Descriptions

In this section, we provide a description for each scenario used in our benchmark. We outline and the reward functions.

**Pitfall**   In this scenario, the agent is located in a very long rectangular corridor. The surface of the area is divided into squares, of which there are 7 in each row. At the beginning of an episode, each of those sections has a 30% chance of being turned into a pit by lowering the height of the floor in the given area. The floor layout is randomized in every episode, so there is no one optimal route to take. Falling into a pit instantly terminates the episode and should be avoided at all costs. Reaching the other end of the corridor also terminates the episode. The agent is tasked to traverse along the corridor as far as possible and potentially reach the other end. Note that there is a $\sim 10^{-5}$ chance that a section of the corridor becomes untraversable due to the stochastic nature of the layout. The agent ought to identify a safe path through the corridor and speedily navigate forwards. The agent can turn left and right, move forward, and accelerate. The agent is proportionally rewarded for how much it has travelled forward in the corridor, and penalized for falling into a pit.

**Arms Dealer**   In this scenario, the agent has to pick up weapons from the ground in a square-shaped room and deliver them to a platform that appears at a random location after a weapon is obtained. The agent can carry multiple (the exact amount is dependent on the weapon) weapons before making the delivery. A new weapon is spawned at a random location whenever one is picked up, i.e., there is always a constant number of weapons on the ground. A new platform is created for every weapon picked up. After a successful delivery, all the platforms disappear, and the agent has to start over by collecting weapons. The agent can turn left and right, move forward, and accelerate. The agent is rewarded for both obtaining a weapon and delivering it. Additionally, a small reward is granted for how much the agent moves in the environment. Finally, the agent is penalized for being idle.

**Hide and Seek**   In this scenario, the agent is randomly spawned in one of 20 possible locations within a maze-like environment. 5 enemies are spawned at random locations at the beginning of an episode. The enemies can only inflict damage at close distance and thus constantly attempt to move closer to the agent. Health kits granting 25 hit points continually spawn in random locations at specified time intervals. The objective of the agent is to survive by hiding from the enemies. The agent should identify the enemies and attempt to constantly move away from them while collecting the health kits when at low health. The agent can turn left and right, move forward, and run. The agent is rewarded for movement, surviving a frame and each health item collected, and penalized for having damage inflicted by enemies.

**Floor is Lava**   In this scenario, the agent is located in a rectangular room divided into 16x16 equal sized squares. The room is filled with lava, which inflicts 1 health point of damage if being stood upon. After every fixed time interval, each square section of lava has a 20% chance of being changed to a platform, which no longer causes damage. The objective is to survive by minimizing the time spent standing in lava. The agent ought to identify the platforms and quickly navigate on top of them as soon as they appear to avoid running out of health. The agent can turn left and right, move forward, and accelerate. The agent is rewarded for surviving a frame, stepping onto a platform whilst having previously stood in lava, and penalized for standing in lava.

**Chainsaw**   In this scenario, the agent is randomly spawned in one of 20 possible locations within a maze-like environment, and equipped with a chainsaw. A fixed number of enemies are spawned at random locations at the beginning of an episode. Additional enemies will continually be added at random unoccupied locations after each time interval. The enemies are rendered immobile, forcing them to remain at their fixed locations. The goal of the agent is to find the enemies, walk up to melee distance from them and saw them in half. The agent can move forward, turn left and right, and use the chainsaw. The agent is granted a reward for moving in the environment and each enemy killed.

**Raise the Roof**   In this scenario, the agent is randomly spawned in one of 20 possible locations within a maze-like environment. The room has a very high ceiling at the beginning of an episode, which, however, starts to be slowly but constantly lowered. At specific locations of the area, there are switches on the walls that can be pressed to raise the ceiling back up a bit. After a switch is pressed,

it disappears and can no longer be used. If the switches are not pressed with a high enough frequency, the ceiling will eventually crush the agent, which terminates the episode. The goal of the agent is thus to locate and press the switches to keep the ceiling high before the episodes timeouts. The agent can move forward, turn left and right, and activate a switch. The agent is rewarded for movement, pressing a switch, and surviving for a frame.

**Run and Gun** In this scenario, the agent is randomly spawned in one of 20 possible locations within a maze-like environment, and equipped with a weapon and unlimited ammunition. A fixed number of enemies are spawned at random locations at the beginning of an episode. Additional enemies will continually be added at random unoccupied locations after each time interval. The enemies are rendered immobile, forcing them to remain at their fixed locations. The goal of the agent is to locate and shoot the enemies. The agent can move forward, turn left and right, and shoot. The agent is granted a reward for each enemy killed.

**Health Gathering** In this scenario, the agent is trapped in a room with a surface, which slowly but constantly decreases the agent's health. Health granting items continually spawn in random locations at specified time intervals. The default health item heals grants 25 hit points. Some environments contain poison vials, which inflict damage to the agent instead of providing health. The objective is to survive. The agent should identify the health granting items and navigate around the map to collect them quickly enough to avoid running out of health. The agent can turn left and right, and move forward. A small reward is granted for every frame the agent manages to survive.

## B Reward Shaping

In this section, we present all components constituting the auxiliary reward function $r$ of each scenario.

**Raise the Roof** $r_t = F + c_1 \cdot s + c_2 \cdot \left\| l_t - l_{t-5} \right\|_2^2$, where $F = 0.01$ is a constant reward granted for surviving an iteration, $s$ is the number of switches pressed, $l$ represents the coordinates of the agent's location, $c_1 = 15$ is the reward of pressing a switch, $c_2 = 0.001$ scales the reward granted for movement, and $t$ is the time step.

**Chainsaw** $r_t = c_1 \cdot k_t + c_2 \cdot \left\| l_t - l_{t-5} \right\|_2^2$, where $k$ is the number of enemies eliminated, $l$ represents the coordinates of the agent's location, $c_1$ is the reward for eliminating an enemy, $c_2 = 0.001$ scales the reward granted for movement, and $t$ is the time step.

**Health Gathering** $r_t = F + c_1 \cdot \mathbb{1}\{h_t > h_{t-1}\}$, where $F = 0.01$ is a constant reward granted for surviving an iteration, $h$ is the remaining health (increases when a health kit is picked up), $c_1 = 15$ scales the reward granted for picking up a health kit, and $t$ is the time step.

**Pitfall** $r_t = d + c_1 \cdot \max(0, x_t - x_{t-1})$, where $d = -1.0$ is the penalty for falling into a pit, $x$ represents the x-axis coordinate of the agent, $c_1 = 0.1$ scales the reward for traversing the corridor, and $t$ is the time step.

**Run and Gun** $r_t = c_1 \cdot k_t + c_2 \cdot \left\| l_t - l_{t-5} \right\|_2^2$, where $k$ is the number of enemies eliminated, $l$ represents the coordinates of the agent's location, $c_1$ is the reward for eliminating an enemy, $c_2 = 0.001$ scales the reward granted for movement, and $t$ is the time step.

**Floor is Lava** $r_t = c_1 \cdot \mathbb{1}\{h_t = h_{t-1} \wedge h_{t-1} < h_{t-2}\} + c_2 \cdot \mathbb{1}\{h_t < h_{t-1}\}$, where $h$ is the agent's health, $c_1 = 3$ is the reward for stepping on a platform from lava, $c_2 = -0.1$ is the penalty for standing on lava, and $t$ is the time step.

**Arms Dealer** $r_t = P + c_1 \cdot w + c_2 \cdot d + c_3 \cdot \left\| l_t - l_{t-5} \right\|_2^2$, where $P = -0.01$ is a constant penalty to avoid passivity, $w$ is the number of weapons acquired, $d$ is the number of weapons delivered, $c_1 = 15$ is the reward of acquiring a weapon, $c_2 = 30$ is the reward for delivering a weapon, $c_3 = 0.001$ scales the reward granted for movement, and $t$ is the time step.

**Hide and Seek** $r_t = F + c_1 \cdot \mathbb{1}\{h_t > h_{t-1}\} + c_2 \cdot \mathbb{1}\{h_t < h_{t-1}\}$, where $F = 0.01$ is a constant reward granted for surviving an iteration, $h$ is the remaining health (increases when a health kit is picked up and decreases when damage is inflicted by an enemy), $c_1 = 5$ is the reward granted for picking up a health kit, $c_2 = -5$ is the penalty for receiving damage from an enemy, and $t$ is the time step. It is relevant to note that picking up a health kit grants more health in comparison to the damage inflicted by the enemy if it were to occur in the same time step, meaning that the agent receives a positive reward.

## C  Framework

In this section, we will discuss our experimental framework in detail.

### C.1  Network Architecture

We use the same convolutional encoder head as in [44] with relu activations, followed by two fully connected layers with 256 neurons each. A Layer Normalization is applied after the first dense layer, with a tanh activation, known to work well with Layer Normalization [4]. The second dense layer is followed by a leaky ReLU activation (with $\alpha = 0.2$). Both the actor and critic networks utilize the presented architecture. The actor network takes an observation as input and generates a Gaussian distribution of action probabilities corresponding to the action space. Each task has its own actor head responsible for producing these outputs. The critic network takes the same input and produces a scalar value for each task head. The network is equipped with multiple heads, each of which is responsible for outputting the action distribution for one of the tasks in the sequence. Their number of output heads is thus equal to the number of tasks in the corresponding sequence. In Appendix **??**, we showcase experiments where only a single head is employed, and the one-hot encoded task id input is provided.

### C.2  Implementation Details

During preprocessing the observation from the environment is downscaled to an $84 \times 84$ pixel RGB image, and 4 of such most recent frames are stacked before fed to the network. At the start of each task, the collected experience is emptied from the replay buffer and the agent uses random policy exploration for 10000 steps to collect experience. The entropy regularization coefficient $\alpha$ is automatically tuned such that the standard deviation of the action distribution on every dimension matches a predefined value $\alpha_{tgt}$. When switching to a new task, we prioritize minimizing forgetting in the actor while allowing the critic to adapt freely. By default, we thus only regularize the actor, except for AGEM, where we found it slightly beneficial to also include the regularizing of the critic network. We use a learning rate decay, which is reset upon the start of a new task.

The agent undergoes sequential training on each task $\mathcal{T}_i$ for a duration of $\Delta = 200K$ iterations each. After every 1000 iterations, we assess the policy's performance on each task within the sequence, conducting 10 episodes for evaluation. A buffer is maintained to store the 50K most recent trajectories, which are subsequently cleared from the buffer at the conclusion of a task. When calculating forgetting, we consider $k = 10$ final performance measurements of a task.

All experiments are conducted on a computer with a 20-core i9-10900K CPU and 2 GeForce RTX 3090 GPUs.

### C.3  Hyperparameters

We adopt most of the hyperparameters from the original SAC algorithm [20] and Continual World [70]. The extensive list is brought in Table 4.

## D  Task Recollection

The individual tasks in our benchmark may vary in overall complexity and exhibit different potential for transfer between each other. In this section, we therefore examine the difficulty of maintaining performance on tasks within the Cross-Objective sequence. Our analysis in Table 5 reveals that the task Pitfall, closely followed by Hide and Seek, exhibit the highest levels of forgetting across

Table 4: Overview of the hyperparameters used in our experiments.

| Hyperparameter | Value |
|---|---|
| *Shared by all algorithms* | |
| optimizer | Adam |
| initial learning rate | $10^{-3}$ |
| learning rate decay | Linear |
| learning rate decay rate | 0.1 |
| learning rate decay steps per task | $10^5$ |
| Adam $\beta_1$, $\beta_2$ | 0.9, 0.999 |
| target action distribution std $\alpha_{tgt}$ | 0.089 |
| hidden layer sizes | $256, 256$ |
| activation convolution | relu |
| activation dense | tanh, lrelu |
| discount ($\gamma$) | 0.99 |
| polyak averaging interpolation factor ($\rho$) | 0.995 |
| frame stack | 4 |
| frame skip | 4 |
| batch size ($n$) | 128 |
| replay buffer size | 50K |
| initial collect steps ($b_{init}$) | $10^4$ |
| steps per task | 200K |
| update frequency (steps) | 5000 |
| gradient steps per update | 50 |
| number of evaluation episodes | 10 |
| regularization number of batches | 10 |
| regularization batch size | 32 |
| *PackNet* | |
| retrain steps | $10^4$ |
| gradient clipping norm | $2 \cdot 10^{-5}$ |
| *MAS* | |
| regularization coefficient | $10^4$ |
| *L2* | |
| regularization coefficient | $10^5$ |
| *EWC* | |
| regularization coefficient | 250 |
| *AGEM* | |
| episodic memory per task | $10^4$ |
| episodic batch size | 128 |
| *VCL* | |
| regularization coefficient | 1 |
| *ClonEx-SAC* | |
| exploration kind | best return |
| regularization coefficient | 100 |
| episodic memory per task | $10^4$ |
| episodic batch size | 128 |

| Scenario | PackNet | MAS | AGEM | L2 | EWC | Fine-tuning | VCL | ClonEx-SAC | Average |
|---|---|---|---|---|---|---|---|---|---|
| Pitfall | **0.18** ± -0.01 | **0.60** ± 0.01 | 0.81 ± 0.06 | **0.04** ± 0.03 | 0.10 ± 0.01 | 0.75 ± 0.09 | 0.55 ± 0.04 | 0.00 ± 0.02 | **0.38** ± 0.31 |
| Arms Dealer | 0.00 ± -0.00 | 0.50 ± -0.14 | 0.63 ± -0.34 | 0.01 ± -0.01 | 0.08 ± -0.06 | 0.47 ± -0.08 | 0.63 ± -0.19 | 0.01 ± -0.01 | 0.29 ± 0.27 |
| Hide and Seek | 0.00 ± -0.01 | 0.25 ± -0.10 | 0.80 ± 0.07 | -0.04 ± 0.07 | **0.14** ± -0.03 | **0.84** ± 0.05 | 0.76 ± -0.01 | **0.06** ± -0.04 | 0.35 ± 0.36 |
| Floor is Lava | -0.18 ± 0.07 | -0.03 ± 0.01 | 0.85 ± -0.04 | 0.01 ± 0.03 | -0.04 ± 0.00 | -0.00 ± 0.03 | 0.64 ± -0.21 | -0.01 ± 0.02 | 0.15 ± 0.35 |
| Chainsaw | -0.01 ± -0.00 | -0.02 ± 0.02 | 0.93 ± -0.01 | -0.08 ± -0.01 | -0.02 ± 0.01 | 0.62 ± -0.08 | 0.61 ± -0.15 | -0.07 ± 0.06 | 0.25 ± 0.38 |
| Raise the Roof | -0.06 ± 0.10 | -0.07 ± -0.02 | **0.97** ± 0.02 | -0.13 ± 0.01 | -0.01 ± -0.02 | 0.46 ± -0.01 | **0.89** ± 0.00 | -0.02 ± 0.03 | 0.25 ± 0.43 |
| Run and Gun | -0.01 ± 0.01 | -0.02 ± 0.01 | 0.93 ± 0.03 | -0.08 ± 0.02 | -0.08 ± -0.02 | 0.20 ± -0.15 | 0.36 ± -0.08 | 0.00 ± -0.00 | 0.16 ± 0.32 |

Table 5: Forgetting of all methods on each individual task of the CO8 sequence, calculated as outlined in Equation 2. The results of tasks that were most forgotten by a given method are highlighted in **bold**. The final task of the sequence is omitted from the comparison, as no forgetting can have occurred.

all methods. However, it is important to note that this comparison may not accurately reflect the inherent difficulty of each task in terms of forgetting.

The ordering of tasks within the sequence can introduce a disparity, as the policy undergoes a greater number of weight updates based on experience gathered from later tasks before completing the entire sequence. This dynamic impacts the difficulty of remembering tasks at the beginning of the sequence. On the other hand, it increases the potential for backward transfer from future tasks, which could potentially enhance performance. However, we did not observe notable instances of backward transfer in our experiments. Consequently, we posit that tasks at the beginning may be slightly disadvantaged in the comparison due to forgetting.

Exploring different orders of sequences would be an interesting avenue for future research. By varying the order, we could better determine the complexity of different scenarios and their impact on performance. This approach would provide more nuanced insights into the difficulty of each task within the sequence. We have provided the functionality in COOM to randomize the tasks in a sequence. However, we have not conducted any experiments in this setting, as a proper evaluation would require numerous trials, which we lack the computational resources for.

Overall, our current analysis highlights the challenges in maintaining performance on the `Pitfall` and `Hide and Seek` tasks. However, further investigation is necessary to fully understand the complexities involved and to identify the most challenging scenarios within the Cross-Objective sequence.

# E Memory and Time Consumption

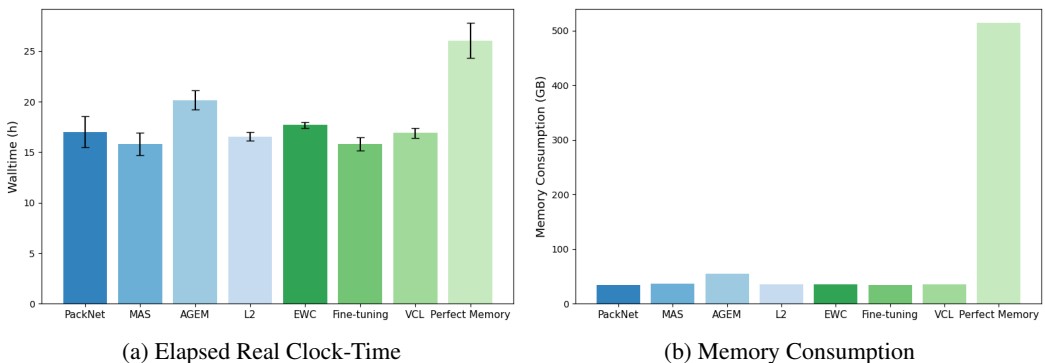

(a) Elapsed Real Clock-Time

(b) Memory Consumption

Figure 7: Time and Memory Consumption on the CD4 task sequence

Our experiments indicate that replay-based methods consume more resources in terms of both time and memory compared to other baselines. Figure 7a depicts the elapsed real clock-time in hours. Most methods indicate an average runtime of 16-17 hours, whereas AGEM and Perfect Memory take around 20 and 25 hours to complete respectively. Figure 7b depicts memory consumption in gigabytes. Most methods exhibit an equal memory consumption of approximately 35GB, as the replay has a fixed size of 50K trajectories. AGEM uses slightly more memory due to storing selective episodic memory from previous tasks. Lastly, Perfect Memory clearly stands out in the comparison

with a consumption of over 500GB. This analysis can be valuable for when resources are a critical factor when selecting an appropriate CL method.

# F    Cross-Objective Challenge

The Cross-Objective Challenge (**COC**) is designed to be a challenge for future methods. In this section we will describe how the task sequence is constructed, and further expand how our baseline methods performed on the sequence. The following list outlines how each scenario is modified to obtain a harder version.

1. **Pitfall**
    - The chance of a tile being a pit is raised from 0.4 to 0.7.
    - The actor property *APROP_Speed* of the agent is decreased from 1.0 to 0.7.

2. **Arms Dealer**
    - The size of the map is increased by 50%.
    - Instead of 5 weapons, only 1 is spawned at a time.
    - Instead of a single weapon type (`Chaingun`), one of 7 weapons (`BFG9000`, `Chaingun`, `Pistol`, `PlasmaRifle`, `RocketLauncher`, `Shotgun`, `SuperShotgun`) is randomly spawned.

3. **Hide and Seek**
    - Number of initially spawned monsters is increased from 10 to 20.
    - Number of initially spawned health kits is decreased from 20 to 10.
    - The actor property *APROP_Speed* of the agent is decreased from 1.0 to 0.7.

4. **Floor is Lava**
    - The change of a tile being a platform is decreased from 0.1 to 0.02.
    - The change interval of the platforms is decreased from 150 to 100 game tics.

5. **Chainsaw**
    - Number of initially spawned monsters is decreased from 20 to 5.
    - The spawn interval of a new monster is decreased from 90 to 30 game tics.
    - Monster health is increased from 1 to 50.

6. **Raise the Roof**
    - The matching texture of the wall sections with switches (`SW1BLUE1`) is changed to (`SW1GRAY1`), so that it no longer differs from the rest of the wall.

7. **Run and Gun**
    - Number of initially spawned monsters is decreased from 20 to 5.
    - The spawn interval of a new monster is decreased from 90 to 30 game tics.
    - Monster health is increased from 1 to 25.

8. **Health Gathering**
    - The map layout from `health_gathering` is replaced with a more complex maze-like version `health_gathering_supreme` from *ViZDoom* [25].
    - The spawn interval of a new health kit is increased from 30 to 60 game tics.

We will henceforth compare results between the CO8 and COC sequences. We depict the training curves of each individual task with its corresponding core success metric in Figure 8. We can observe a slight performance increase in each task on COC, but it significantly falls below the CO8 counterparts. Figure 9 indicates that PackNet, the best method in our previous benchmark evaluations, also stands out in COC. The continual evaluation results in Figure 10 further show how all displayed methods underperform on the hard COC.

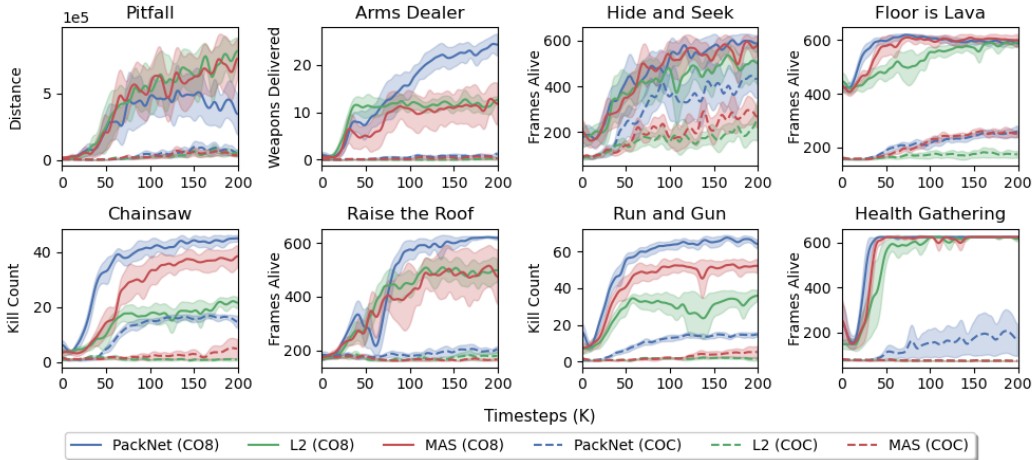

Figure 8: Performance comparison of PackNet and MAS on individual tasks of CO8 and their complex counterparts in COC. Each graph only depicts the time steps during which the task was trained. COC remains unsolved for MAS and even PackNet, which was the best performing method on other sequences.

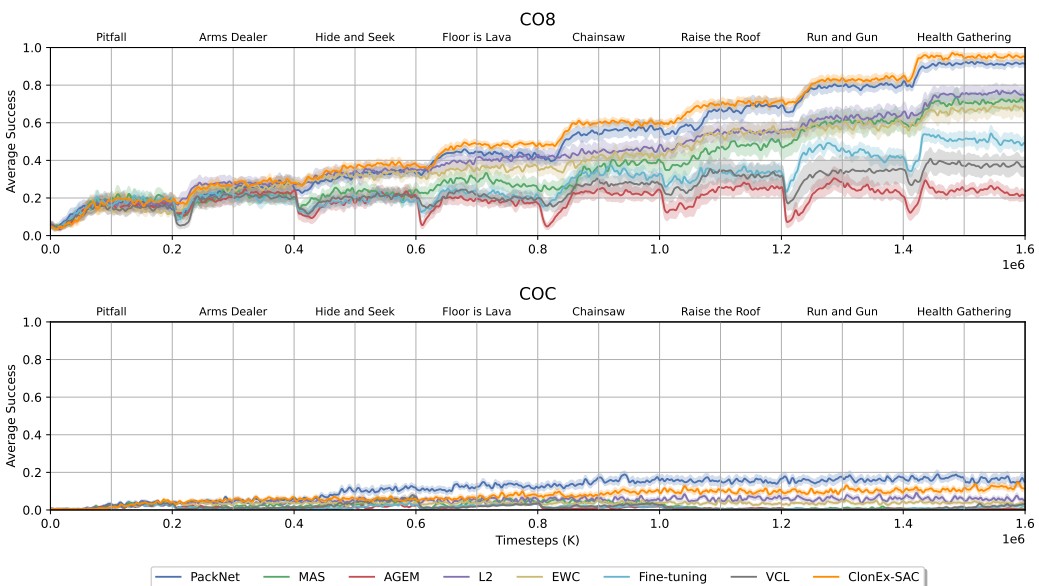

Figure 9: Comparison of average success between CO8 and COC.

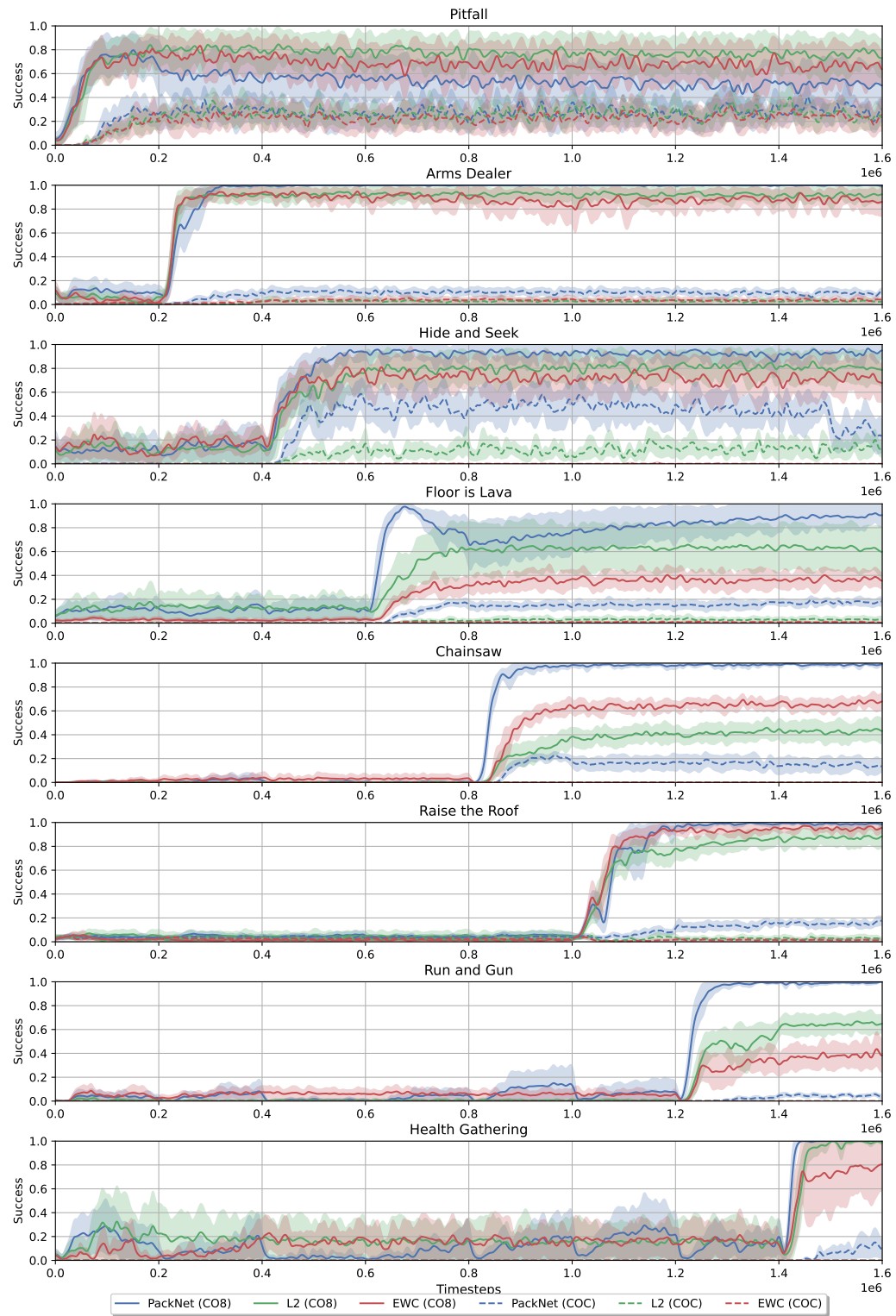

Figure 10: Continual evaluation results of select methods on CO8 and COC.

# G   Sequence Length Comparison

Another way of assessing forward transfer would be to compare the average performance on a 4-task sequences with the corresponding $2^{nd}$ half of its 8-task counterpart. The tasks and their order in this comparison are the same. The only difference here is that the 8-task sequence has had the potential to learn relevant competencies from previous tasks in its $1^{st}$ half. If the $2^{nd}$ half of the 8-task sequence has higher performance then this would be an indicator that knowledge from the initial tasks has been transferred forward to the later

Table 6: Average performance comparison of 4-task sequences with the $2^{nd}$ half of the corresponding 8-task sequence.

| Method | CD4 | CD8$^{2nd}$ | CO4 | CO8$^{2nd}$ |
|---|---|---|---|---|
| **PackNet** | 0.92 ± 0.04 | 0.92 ± 0.03 | 0.87 ± 0.13 | 0.91 ± 0.04 |
| **MAS** | 0.55 ± 0.33 | 0.83 ± 0.16 | 0.72 ± 0.15 | 0.76 ± 0.23 |
| **AGEM** | 0.35 ± 0.24 | 0.43 ± 0.18 | 0.42 ± 0.25 | 0.35 ± 0.32 |
| **L2** | 0.71 ± 0.25 | 0.71 ± 0.12 | 0.80 ± 0.16 | 0.54 ± 0.14 |
| **VCL** | 0.46 ± 0.26 | 0.43 ± 0.17 | 0.40 ± 0.28 | 0.61 ± 0.26 |
| **Fine-tuning** | 0.59 ± 0.23 | 0.72 ± 0.18 | 0.50 ± 0.19 | 0.69 ± 0.21 |
| Average | 0.595 | **0.673** | 0.618 | **0.643** |

ones. We present this comparison in Table 6. Across all methods, we report a **4.04%** performance increase from CO4 to CO8$^{2nd}$ and **12.8%** from CD4 to CD8$^{2nd}$. A performance decrease only occurred in 3/12 of cases. We can indeed thus conclude that the $1^{st}$ half of 8-task sequences fastens learning the $2^{nd}$ half. We posit that the **C**ross-**D**omain setting has higher transfer due to the shared objective across tasks, meaning that learned behaviour can better bolster performance in further tasks.

# H   Network Plasticity Experiments

In our experiments for network plasticity, we halve the number of steps for each task, reducing them to 100K, as we found it to be sufficient for convergence. The entire sequences thus lasts 4M and 8M environment iterations in CO4 and CO8 respectively. After a task is finished, we maintain the network weights and only reset the learning rate decay and replay buffer. Instead of measuring Success, we assess the individual performance metric of each task. We wish to observe whether the initial performance achieved by the agent on a given task deteriorates over successive trials when re-encountering it. We additionally depict the learning curves of the SAC agent on the CO8 task sequence in Figure 11.

Similar to the results on CO4 in the main paper (Figure 4), the attained peak performance remains stable throughout the iterations. However, on CO8, we can witness a discernible dip in performance across most tasks midway through the entire sequence, yet notably, this decline is followed by a resurgence towards the end. Further, on CO4 we can observe that the initial iteration of some tasks is

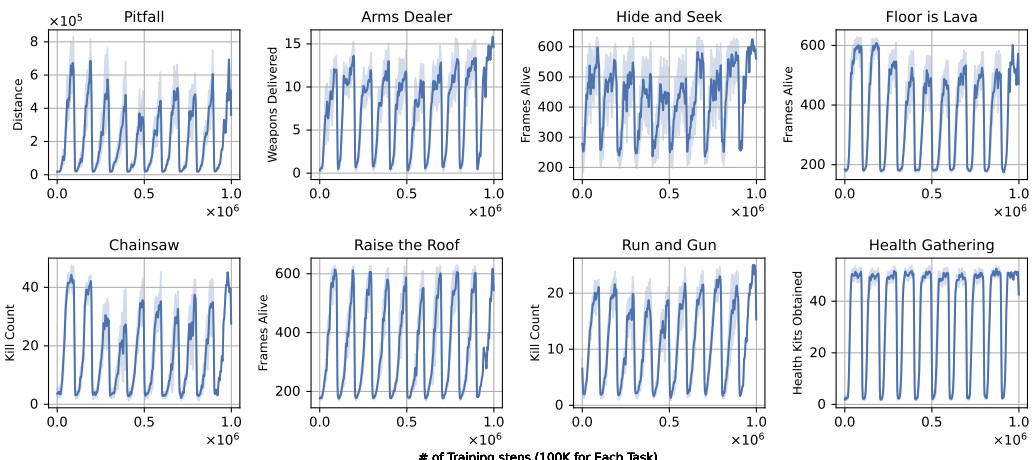

Figure 11: **SAC does not exhibit a loss of network plasticity.** Measuring the individual performance metric of each task across 5 seeds displays that repeated exposure to the same tasks throughout 10 iterations of the CO8 sequence does not hinder the capacity to reattain peak performance.

substantially lower than the subsequent runs. This, however, does not appear for the same tasks on CO8, because the agent has already learned to succeed in other tasks and is capable of transferring knowledge forward.

# I    Method Variations

Table 7 presents the full results of incorporating variations to our baseline methods. Figures 12 and 13 depict the average success curves of selects methods on the CO8 and CO4 task sequences respectively. PackNet stands the strongest in terms of stability across the variations, as not a single variations causes a substantial degradation of performance. Regularization-based methods stand out in our evaluations as the most volatile, as their performance is very likely to completely degrade when employed with variations. The variation that improved the performance of most methods is the *Noise* augmentation.

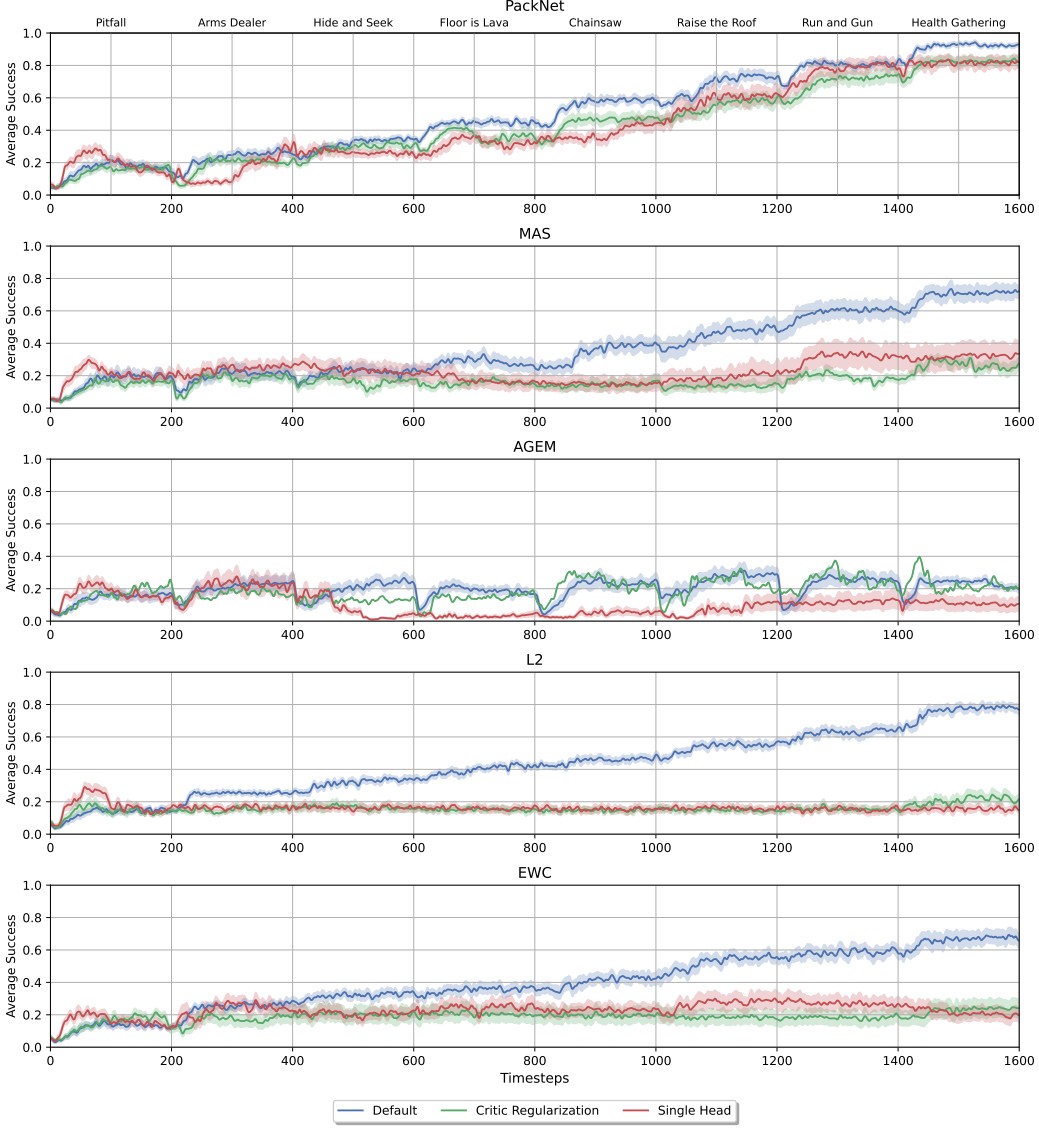

Figure 12: Performance comparison on the CO8 sequence of select methods between critic regularization and single output with the original setting.

Table 7: Method variation performance on CO8. Results higher than the default are highlighted in **bold**.

| Method | Default | Conv | Shift | Noise | LSTM | Single Head | Critic Reg |
|---|---|---|---|---|---|---|---|
| PackNet | $0.84_{\pm0.07}$ | $0.72_{\pm0.09}$ | $\mathbf{0.87}_{\pm0.04}$ | $\mathbf{0.88}_{\pm0.03}$ | $0.83_{\pm0.06}$ | $0.68_{\pm0.15}$ | $0.70_{\pm0.12}$ |
| MAS | $0.58_{\pm0.17}$ | $0.08_{\pm0.07}$ | $0.48_{\pm0.15}$ | $0.47_{\pm0.15}$ | $0.01_{\pm0.01}$ | $0.29_{\pm0.18}$ | $0.25_{\pm0.12}$ |
| AGEM | $0.30_{\pm0.13}$ | $0.02_{\pm0.00}$ | $\mathbf{0.40}_{\pm0.09}$ | $\mathbf{0.33}_{\pm0.02}$ | $\mathbf{0.48}_{\pm0.09}$ | $0.12_{\pm0.11}$ | $0.28_{\pm0.04}$ |
| L2 | $0.73_{\pm0.10}$ | $0.34_{\pm0.05}$ | $\mathbf{0.75}_{\pm0.11}$ | $\mathbf{0.79}_{\pm0.08}$ | $0.01_{\pm0.01}$ | $0.28_{\pm0.10}$ | $0.28_{\pm0.10}$ |
| VCL | $0.39_{\pm0.15}$ | $0.05_{\pm0.04}$ | $0.30_{\pm0.10}$ | $0.29_{\pm0.13}$ | $0.01_{\pm0.00}$ | $0.31_{\pm0.08}$ | $\mathbf{0.41}_{\pm0.06}$ |
| ClonEx-SAC | $0.89_{\pm0.07}$ | $0.26_{\pm0.01}$ | $0.89_{\pm0.00}$ | $\mathbf{0.90}_{\pm0.06}$ | $0.02_{\pm0.01}$ | $0.79_{\pm0.11}$ | $0.87_{\pm0.04}$ |
| EWC | $0.65_{\pm0.13}$ | - | - | - | - | $0.32_{\pm0.15}$ | $0.26_{\pm0.17}$ |
| Fine-tuning | $0.44_{\pm0.11}$ | - | - | - | - | $0.13_{\pm0.13}$ | - |

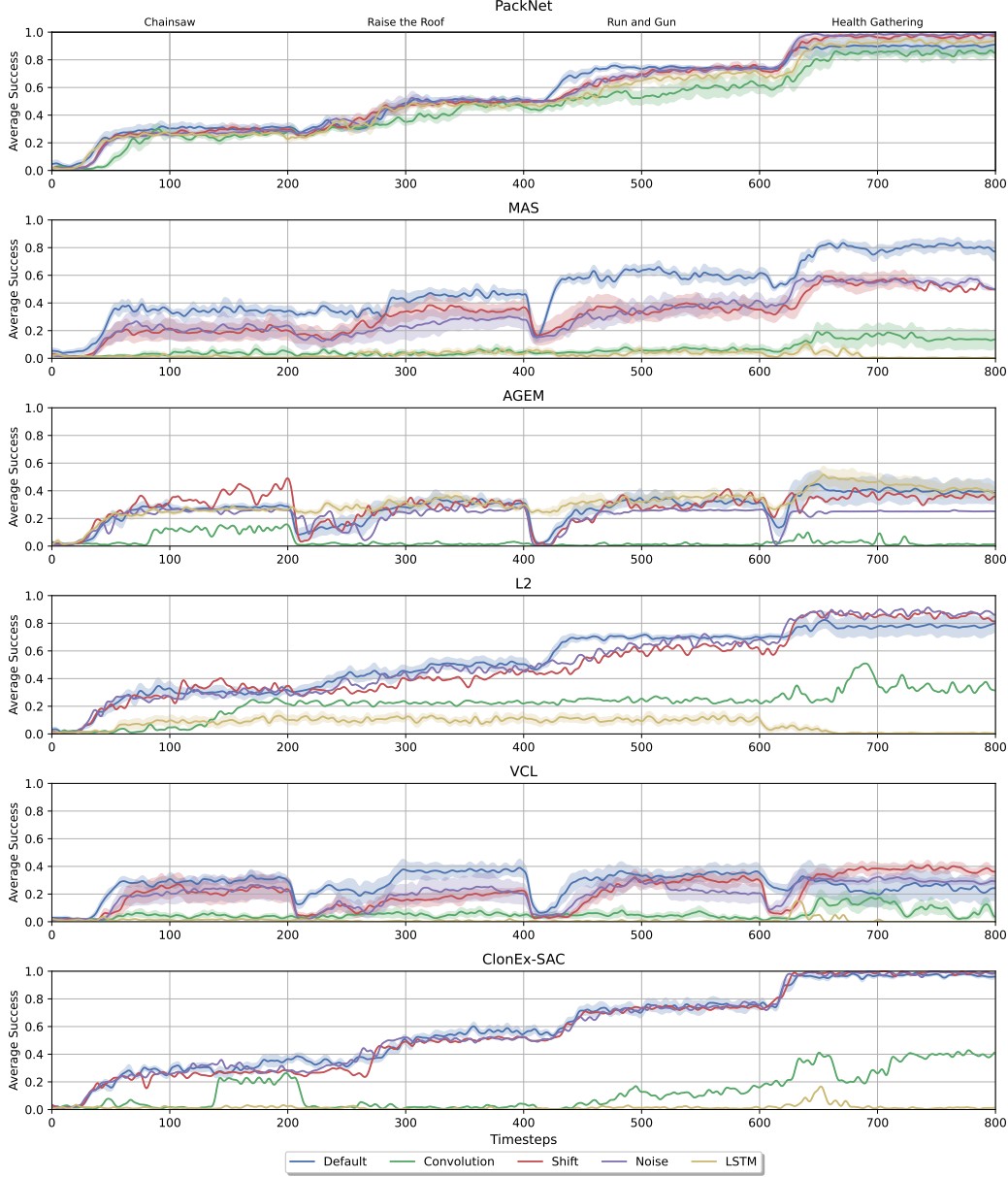

Figure 13: Performance comparison on the CO4 sequence of select methods between image augmentation methods and and LSTM encoder with the original setting.

## J    Repeated Sequences

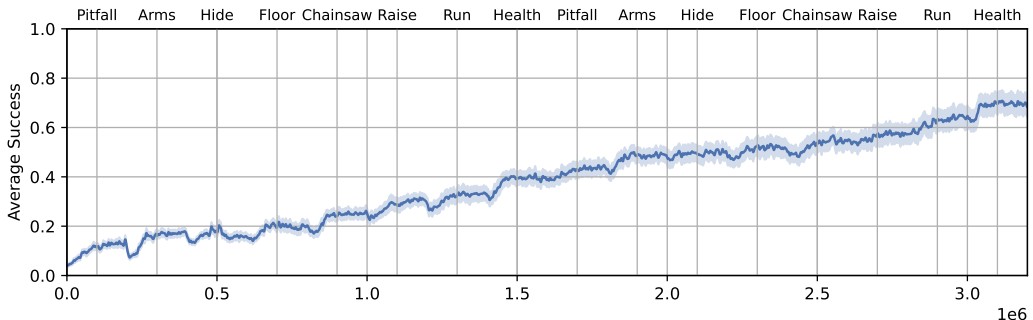

Figure 14: Average cumulative success off PackNet on CO16.

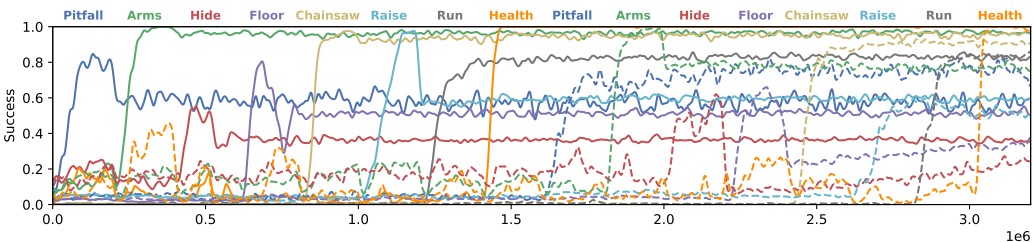

Figure 15: Continual evaluation of PackNet on each task of CO16.

In line with the approach used in [70], we have repeated our core sequences to generate 16-task sequences for both modalities. More tasks present a greater challenge in retaining previously acquired knowledge. Additionally, it enables a thorough evaluation of how methods leverage their acquired competencies when encountering previously learned tasks anew. Given the substantial computational cost associated with conducting experiments on the 16-task sequences, we have limited our evaluation to PackNet applied to CO16. We present a visual overview of the results in Figures 14 and 15.

## K    Cross-Domain Sequence

In this section, we provide a detailed description of the modifications implemented to compose the Cross-Domain sequences based on the default environment of the `Run and Gun` scenario. The specific modification types for each environment are displayed in Table 8.

Table 8: Environment modification types of *Run and Gun*

| Environment | Obstacles | Green | Resized | Monsters | Default | Red | Blue | Shadows |
|---|---|---|---|---|---|---|---|---|
| **Modification Type** | Decorations | Textures | Target & Agent Size | Target Type | - | Textures | Textures | Target Rendering |

**Decorations:** This modification involves adding various in-game elements (e.g., torches, pillars, barrels, lamps), to the game environment at fixed locations. These decorations serve as 1) distractions by having the agent considering them relevant, and 2) obstacles by impeding the agent's movement.

**Textures:** The textures used for various game assets, such as walls, floors, and ceilings, are altered in this modification. Different textures are applied to create distinct visual appearances for each task, further increasing the visual disparity across tasks.

**Target & Agent Size:** This modification involves adjusting the width and height of targets and the height of the agent within the game environment. Smaller or narrower targets can be more difficult to detect. Changing the height of the agent affects the view height and thus creates a visual shift of the perceived environment.

**Target Type:** In this modification, the visual appearance of targets is altered. We use various enemies from the original game and render them immobilized.

**Target Rendering:** This modification focuses on changing the visual rendering of the targets. For example, targets can be rendered with different visual effects or styles, making them visually more distinct or harder to detect and requiring the agent to adapt its perception and recognition abilities.

To implement these modifications, we utilize the SLADE3 tool, which enables us to customize various aspects of the DOOM game environment. Each modification type contributes to the visual disparity and complexity across tasks, challenging the agent's perception, decision-making, and adaptation capabilities, creating a comprehensive continual learning task sequence. In the following we provide a more comprehensive description of changes.

1. **Obstacles**
   - A total of 25 elements named *Tall Blue Torch*, *Tall Green Torch*, and *Tall Red Torch* with respective in-game id's [44, 45, 46] are constructed at fixed locations of the map, which act as obstacles

2. **Green**
   - Wall texture *A-CAMO1*
   - Floor texture *AQF022*
   - Ceiling texture *DOGRMSC*

3. **Resized**
   - The actor properties *APROP_ScaleX* and *APROP_ScaleY* of enemies are randomized in the range of [0.3, 3.0]
   - The actor property *APROP_ViewHeight* of the agent is randomized in the range of [0.0, 70.0]

4. **Monsters**
   - Replaced targets with immobile enemies: *Arachnotron*, *Fatso*, *Cacodemon*, *Archvile*, *Revenant*
   - The number of initially spawned enemies is set to 3 per type
   - The actor property *APROP_Invulnerable* of the agent is set to *True*

5. **Default**
   - The map layout originates from the *ViZDoom* [25] scenario *health_gathering_supreme*
   - Wall textures *ICKWALL3*
   - Floor texture *AQF051*
   - Ceiling texture *FLAT19*
   - The surface no longer inflicts damage to the agent
   - Stationary targets are spawned at random locations around the map
   - The target type is *CommanderKeen*
   - The health of the target is set to 1
   - The number of initially spawned targets is set to 20
   - The spawn delay of every subsequent target is set to 30 game ticks
   - The agent is equipped with a pistol and 200 bullets

6. **Red**
   - Wall texture *FIREWALL*
   - Floor texture *CRACKLE4*
   - Ceiling texture *DORED*

7. **Blue**
   - Wall texture *FIREBLU2*
   - Floor texture *FLAT14*
   - Ceiling texture *CEIL4_3*

8. **Shadows**
   - The actor property *APROP_RenderStyle* of targets is set to *STYLE_Shadow*

# L Limitations

**Sequence ordering**    In our benchmark, we focused on evaluating performance and transferability in the context of a fixed sequence ordering. We acknowledge that varying the order of the sequences could introduce additional dynamics and complexities to the continual learning (CL) process.

Different sequence orderings have the potential to impact the learning process by introducing variations in task difficulty, temporal dependencies, or interference patterns. Exploring multiple sequence orderings could provide valuable insights into the robustness and adaptability of CL methods across different scenarios.

However, due to practical constraints, such as limited computational resources and time, we were unable to thoroughly investigate the effects of sequence ordering in our experiments. Consequently, our benchmark considers only a single predefined ordering of the sequences.

Future research endeavors could delve into the impact of sequence ordering by systematically exploring different orderings and analyzing their effects on CL performance. This would enable a more comprehensive understanding of the challenges and opportunities associated with varying sequence orderings in continual learning scenarios.

**Restricted Action Space**    In order to simplify the training process, enhance ease of learning, and bring more focus to the continual learning paradigm, rather than very complex RL challenges, we opted to restrict the total action space of the DOOM video game. By doing so, we only included actions that we deemed essential for effectively achieving the established objectives.

In addition to maintaining the core actions required for navigating within the game environment, we kept a single additional action called for each scenario, which we referred to as EXECUTE. For instance, in some scenarios, the action resulted in the agent accelerating, while in others, it corresponded to firing the weapon. This approach allowed us to condense the action space even further, particularly in cross-objective sequences where certain actions would have otherwise been rendered unnecessary or redundant in some tasks.

However, this potentially confuses the agent both when selecting the best action during training, but also, when attempting to maintain performance on learned tasks, in which performing an action on a previous task from the given space would result in a different action in-game compared to the one in the task being trained. On the other hand, allowing the agent to have a broader range of actions at its disposal may have the potential to enhance overall performance, especially when employing more sophisticated and powerful methods. A wider variety of actions would enable the agent to explore different strategies and potentially discover more effective approaches to achieving the objectives.

**Sequence length**    The number of tasks in our benchmark is relatively small, and they are not repeated. Introducing task repetition in a continual learning (CL) benchmark can yield additional interesting observations. However, due to the extensive runtime requirements for low-level budget setups, we opted to avoid long sequences in this benchmark.

The dimensionality of the observation space and the rendering of the environment contribute to longer learning times for a single task. We established a fixed task length of 200K timesteps, as our observations indicated that this was sufficient for reaching convergence. In our limited experiments with longer task lengths, we found that no method exhibited notable performance improvement after this number of iterations nor managed to learn anything meaningful beyond this point.

While longer task lengths and task repetition could provide valuable insights, the constraints of our experimental setup, including limited resources and time, prevented us from exploring these aspects in-depth. However, future studies with more resources and computational power could investigate these factors and potentially uncover further insights in the context of continual learning benchmarks.

**Task boundaries**    In our benchmark, we adopted the use of explicit task boundaries to define the transitions between different tasks. While this approach provides a clear and well-defined setting for evaluation, it does come with certain limitations.

By relying on explicit task boundaries, we assume that the learner has prior knowledge or external cues to identify and differentiate between tasks. In real-world scenarios, however, the presence of explicit task boundaries may not be readily available or easily discernible. Therefore, extending

the benchmark to incorporate task inference mechanisms would be a valuable direction for future research.

Task inference mechanisms enable the learner to autonomously detect task boundaries without explicit supervision. Incorporating such mechanisms in the benchmark would allow for a more comprehensive evaluation of CL algorithms that possess the capability to infer task transitions.

Additionally, the current format of the benchmark does not explicitly test algorithms' ability to handle continuous distributional drift. The absence of continuous drift scenarios limits the assessment of CL methods in dynamic and evolving environments. Future iterations of the benchmark could include settings that introduce continuous distributional drift, thereby enabling the evaluation of algorithms that can adapt to changing data distributions over time.

## M Action Distributions

Apart from visualizing the actions executed during training, it is more relevant for the CL conundrum to observe how a particular method manages to preserve the obtained distribution after training on a particular task has concluded and the sequence continues with new tasks being introduced. To this end, we depict the distribution of actions continuously performed by PackNet and L2 on each task within the CO8 task sequence in Figures 16 and 17.

Given the experimental results in Table 3, we know that PackNet and L2 significantly outperformed VCL in both average performance and forgetting. Observing the figures on action distributions, this manifests in VCL being noticeably incapable of maintaining the obtained distribution as new tasks come along, whereas PackNet and L2 have minimal fluctuation after the policy has been adjusted for a given task. We can notice that although PackNet and L2 both had a relatively high performance on the task sequence, the distributions for particular tasks can drastically vary. This suggests that there are a multitude of possible policies for acquiring adequate results on a task. For instance, there are two clearly dominating actions which PackNet utilizes in *Arms Dealer*, whereas L2 has more variety in the selection of actions, with MOVE_FORWARD being the most used one. Interestingly, L2 also has low variance in the distribution before a task has been encountered, whereas for PackNet it is more random. The L2 regularization after the first task seems to fix the distribution also for all other future tasks.

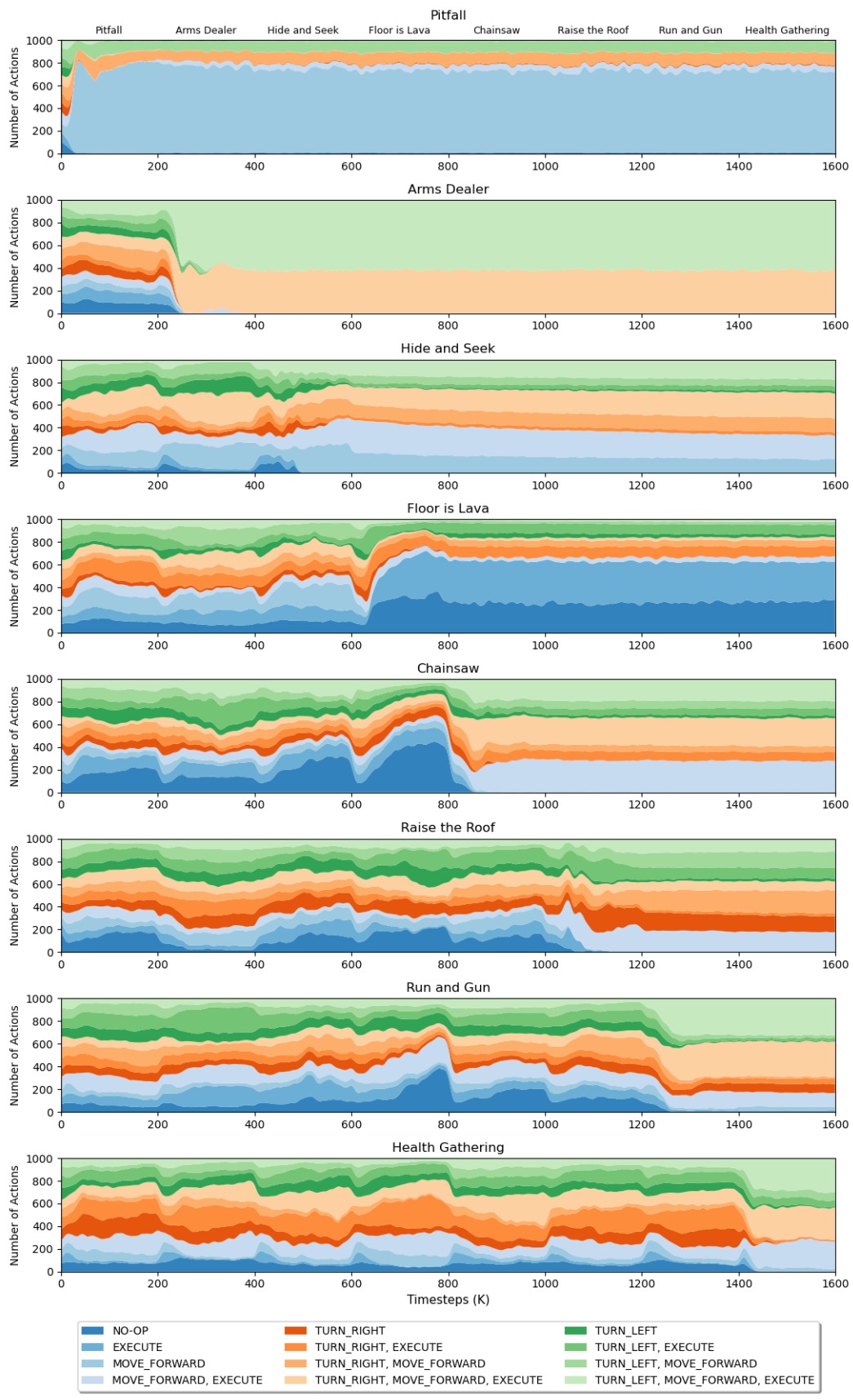

Figure 16: Actions executed by **PackNet** on CO8.

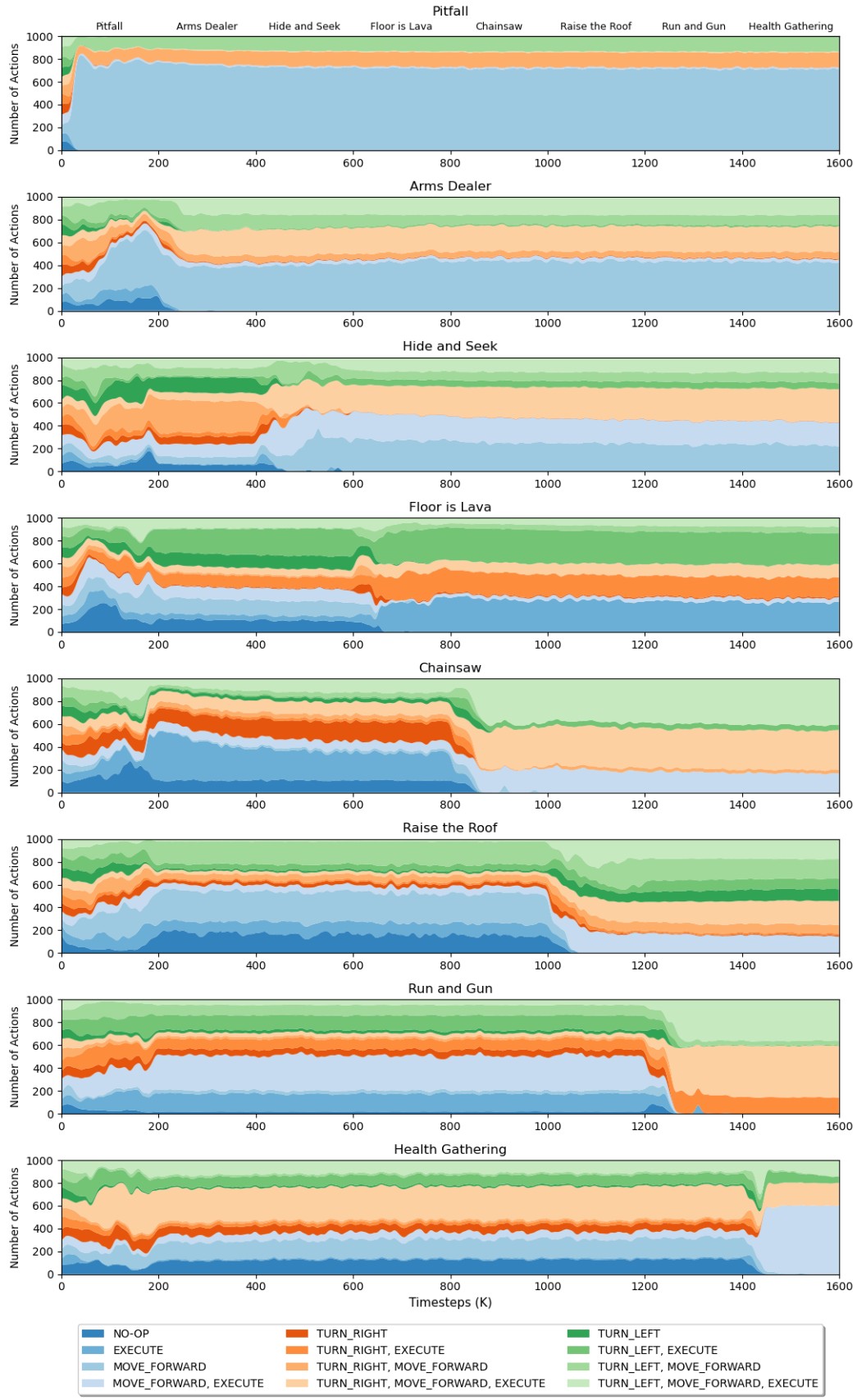

Figure 17: Actions executed by **L2** on CO8.

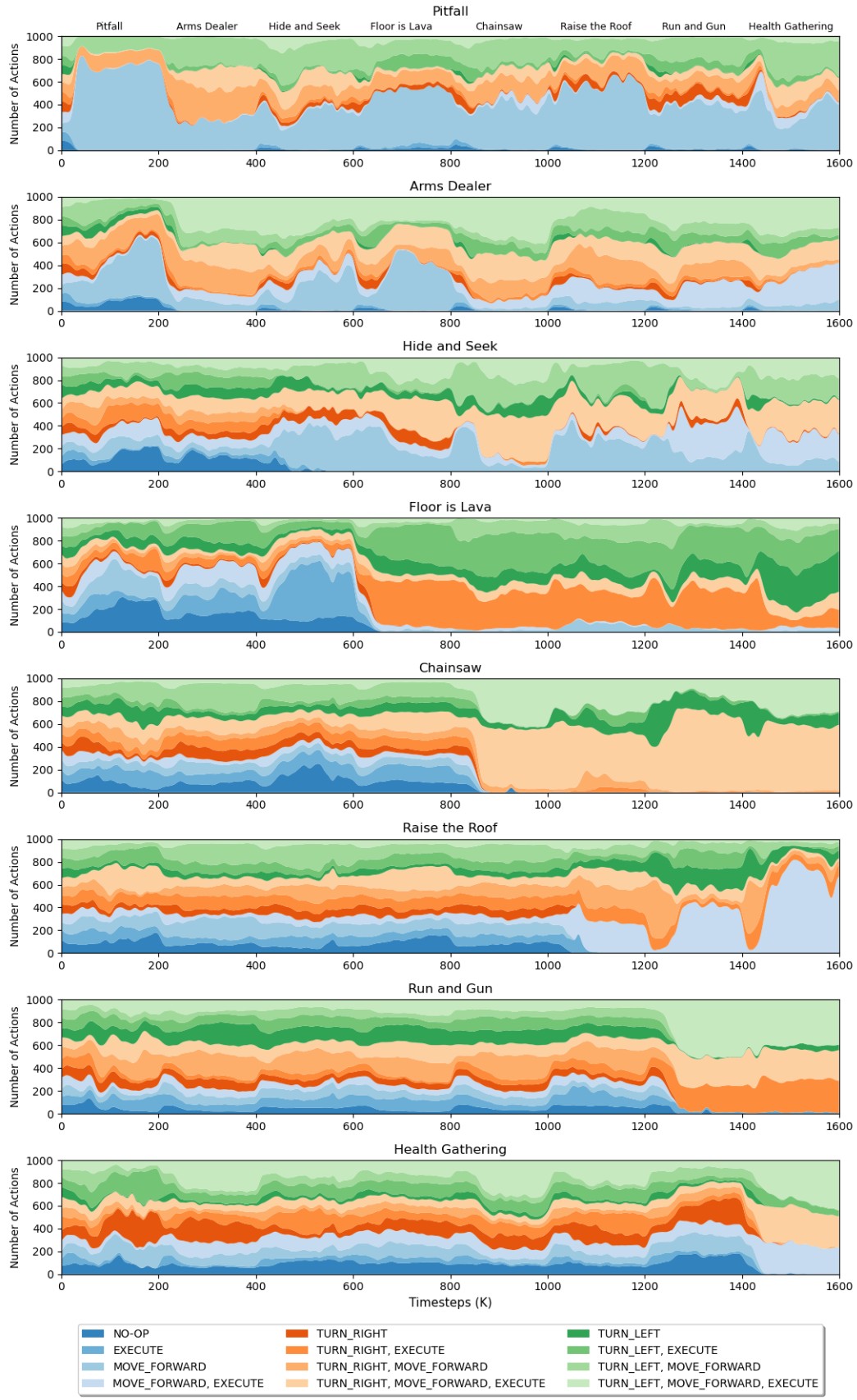

Figure 18: Actions executed by **VCL** on CO8.

# N  Extended Results and Plots

In Table 19 we present the main experimental results with 95% confidence intervals.

| Sequence | PackNet | MAS | AGEM | L2 | EWC | VCL | Fine-tuning | ClonEx-SAC | Perfect Memory |
|---|---|---|---|---|---|---|---|---|---|
| CD4 | **0.92** ±0.02 | 0.55 ±0.17 | 0.35 ±0.12 | 0.71 ±0.13 | 0.69 ±0.08 | 0.46 ±0.14 | 0.59 ±0.12 | 0.87 ±0.03 | 0.89 ±0.05 |
| CO4 | 0.87 ±0.07 | 0.72 ±0.08 | 0.42 ±0.13 | 0.80 ±0.08 | 0.65 ±0.11 | 0.40 ±0.14 | 0.50 ±0.10 | 0.86 ±0.04 | **0.89** ±0.06 |
| CD8 | 0.91 ±0.05 | 0.82 ±0.12 | 0.30 ±0.10 | 0.87 ±0.06 | 0.76 ±0.11 | 0.36 ±0.13 | 0.45 ±0.11 | **0.92** ±0.03 | - |
| CO8 | 0.82 ±0.09 | 0.58 ±0.17 | 0.30 ±0.13 | 0.71 ±0.13 | 0.65 ±0.13 | 0.39 ±0.18 | 0.44 ±0.11 | **0.89** ±0.07 | - |
| COC | **0.20** ±0.08 | 0.04 ±0.02 | 0.02 ±0.01 | 0.10 ±0.05 | 0.07 ±0.04 | 0.03 ±0.02 | 0.02 ±0.01 | 0.13 ±0.05 | - |
| Average | **0.74** ±0.06 | 0.54 ±0.11 | 0.28 ±0.10 | 0.64 ±0.09 | 0.56 ±0.09 | 0.33 ±0.12 | 0.40 ±0.09 | 0.73 ±0.05 | - |

(a) Performance

| Sequence | PackNet | MAS | AGEM | L2 | EWC | VCL | Fine-tuning | ClonEx-SAC | Perfect Memory |
|---|---|---|---|---|---|---|---|---|---|
| CD4 | 0.00 ±0.00 | 0.50 ±0.24 | 0.80 ±0.06 | **-0.02** ±0.00 | 0.00 ±0.01 | 0.77 ±0.08 | 0.63 ±0.19 | 0.00 ± 0.00 | -0.01 ±0.01 |
| CO4 | 0.01 ±0.01 | 0.24 ±0.13 | 0.80 ±0.22 | 0.00 ±0.00 | 0.04 ±0.05 | 0.82 ±0.09 | 0.71 ±0.06 | **-0.03** ±0.02 | 0.02 ±0.01 |
| CD8 | 0.01 ±0.00 | 0.14 ±0.12 | 0.86 ±0.02 | -0.03 ±0.03 | **-0.04** ±0.03 | 0.73 ±0.10 | 0.75 ±0.07 | -0.03 ±0.02 | - |
| CO8 | -0.01 ±0.02 | 0.17 ±0.03 | 0.84 ±0.03 | **-0.04** ±0.02 | 0.02 ±0.02 | 0.64 ±0.09 | 0.48 ±0.02 | 0.00 ±0.01 | - |
| COC | 0.04 ±0.00 | 0.09 ±0.03 | 0.11 ±0.05 | **0.00** ±0.01 | **0.00** ±0.00 | 0.16 ±0.06 | 0.10 ±0.04 | 0.01 ±0.00 | - |
| Average | 0.01 ±0.00 | 0.23 ±0.10 | 0.68 ±0.03 | **-0.02** ±0.01 | 0.00 ±0.01 | 0.62 ±0.06 | 0.53 ±0.06 | -0.01 ±0.01 | - |

(b) Forgetting

| Sequence | PackNet | MAS | AGEM | L2 | EWC | VCL | Fine-tuning | ClonEx-SAC | Perfect Memory |
|---|---|---|---|---|---|---|---|---|---|
| CD4 | **0.40** ±0.08 | 0.11 ±0.32 | 0.10 ±0.21 | -0.28 ±0.32 | -0.41 ±0.10 | 0.21 ±0.29 | 0.32 ±0.19 | 0.11 ±0.05 | 0.30 ±0.15 |
| CO4 | -0.24 ±0.32 | -0.04 ±0.06 | **0.03** ±0.10 | -0.60 ±0.37 | -0.77 ±0.13 | -0.57 ±0.30 | -0.01 ±0.15 | -0.26 ±0.09 | 0.03 ±0.20 |
| CD8 | 0.19 ±0.25 | 0.25 ±0.03 | 0.17 ±0.21 | 0.07 ±0.05 | -0.55 ±0.16 | 0.04 ±0.08 | **0.28** ±0.11 | 0.13 ±0.05 | - |
| CO8 | 0.25 ±0.05 | 0.01 ±0.10 | 0.23 ±0.02 | -0.32 ±0.07 | -0.38 ±0.13 | 0.20 ±0.06 | 0.23 ±0.04 | **0.27** ±0.10 | - |
| COC | **0.08** ±0.01 | 0.02 ±0.00 | 0.02 ±0.00 | -0.00 ±0.01 | -0.01 ±0.00 | 0.05 ±0.02 | 0.02 ±0.00 | 0.03 ±0.01 | - |
| Average | 0.13 ±0.14 | 0.07 ±0.10 | 0.11 ±0.11 | -0.23 ±0.16 | -0.43 ±0.10 | -0.01 ±0.15 | **0.17** ±0.10 | 0.06 ±0.06 | - |

(c) Transfer

Figure 19: Extended results of average performance, forgetting and forward transfer with 95% confidence intervals across 10 seeds. The result of the best performing method is highlighted in bold. In the case of ties, the one with the highest confidence in selected.

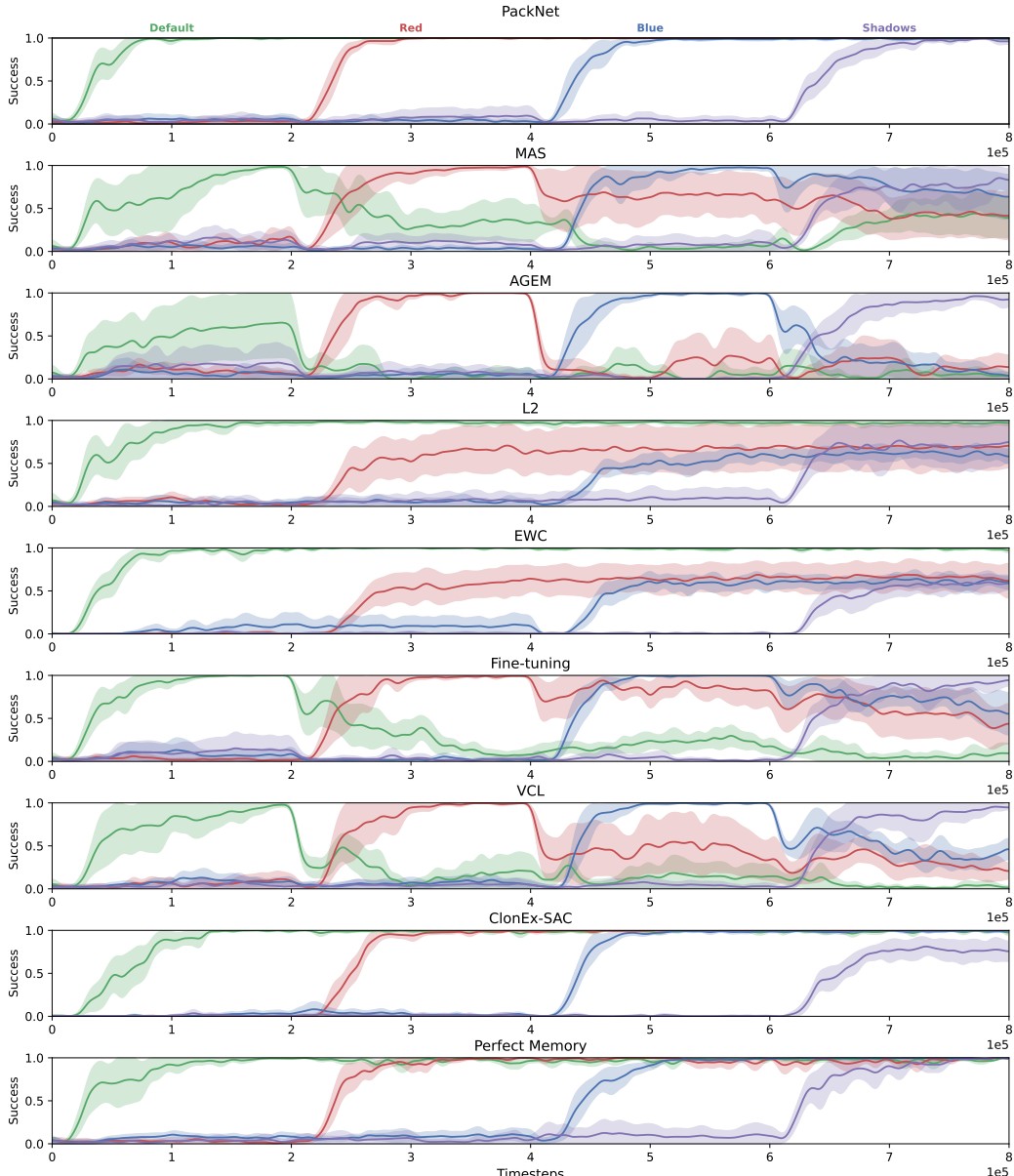

Figure 20: Continual evaluation results on the CD4 sequence.

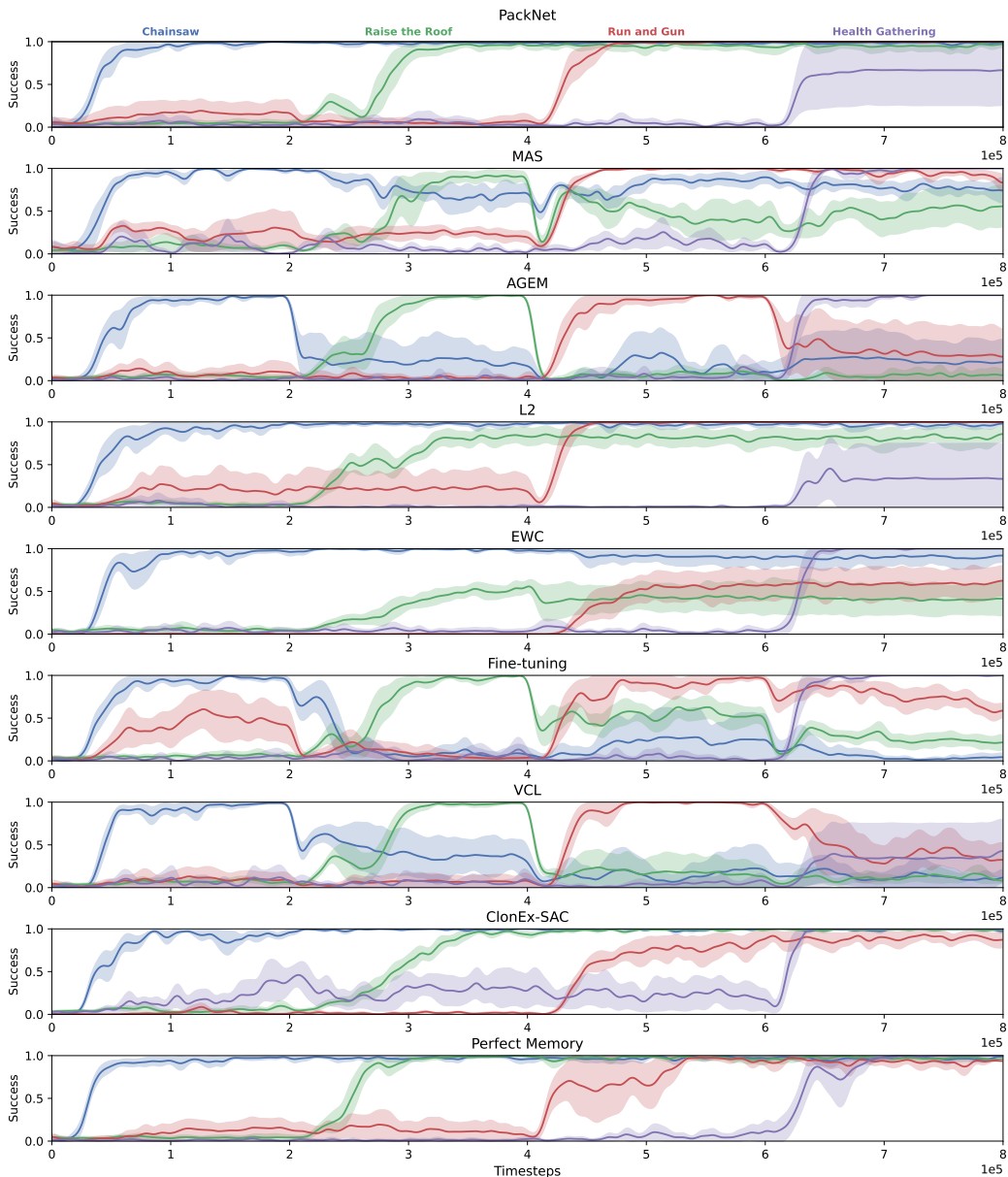

Figure 21: Continual evaluation results on the CO4 sequence.

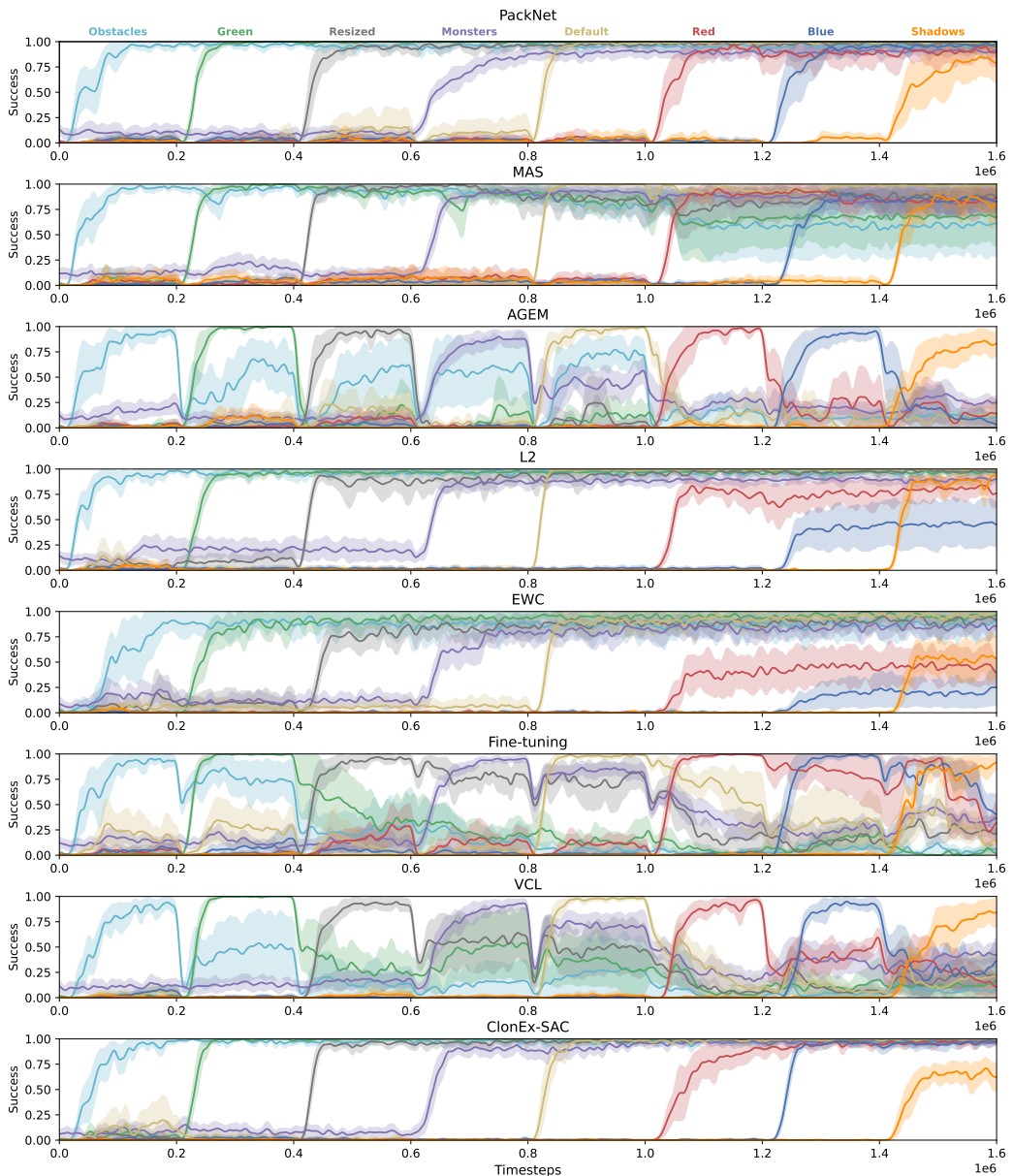

Figure 22: Continual evaluation results on the CD8 sequence.

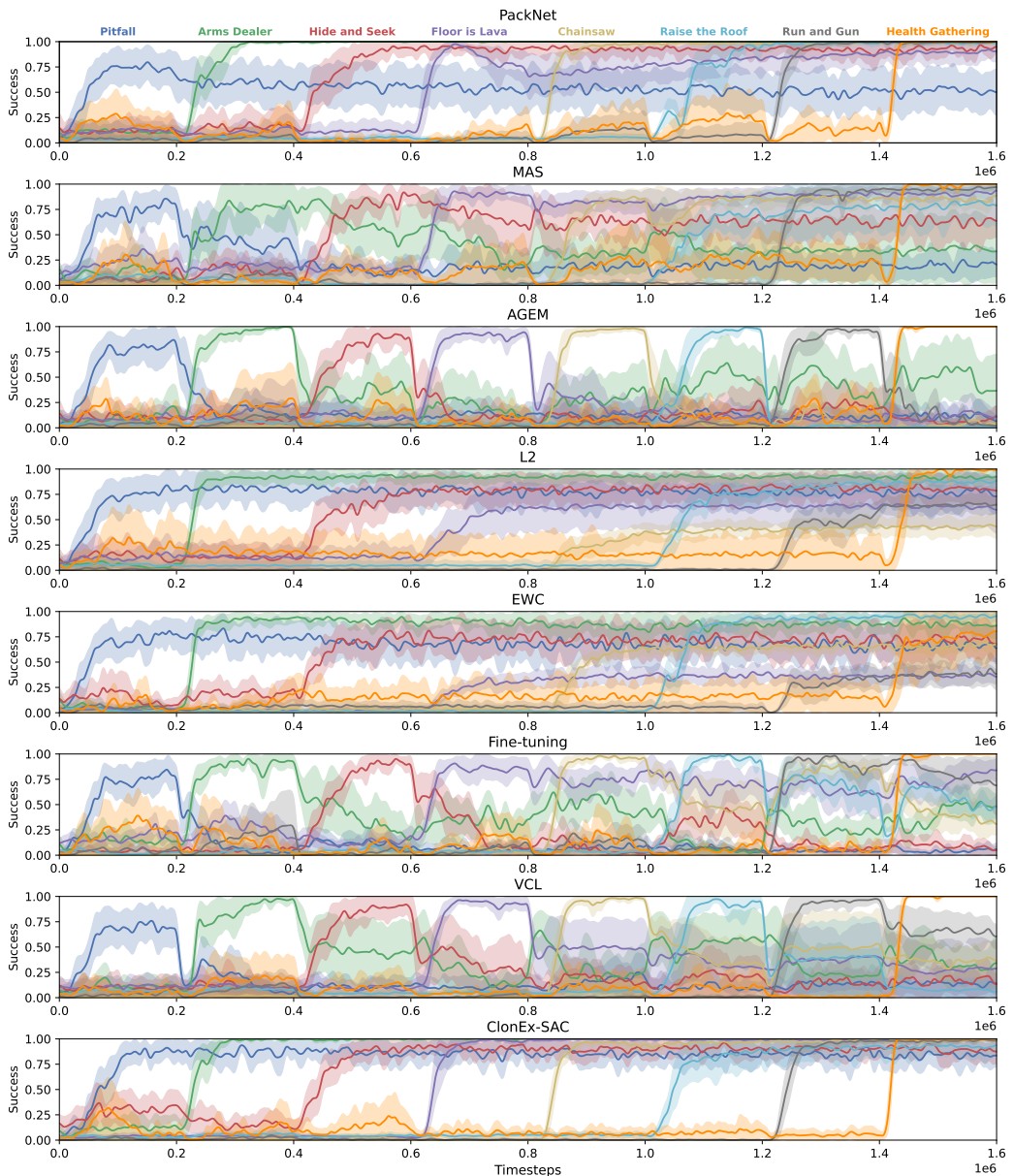

Figure 23: Continual evaluation results on the CO8 sequence.

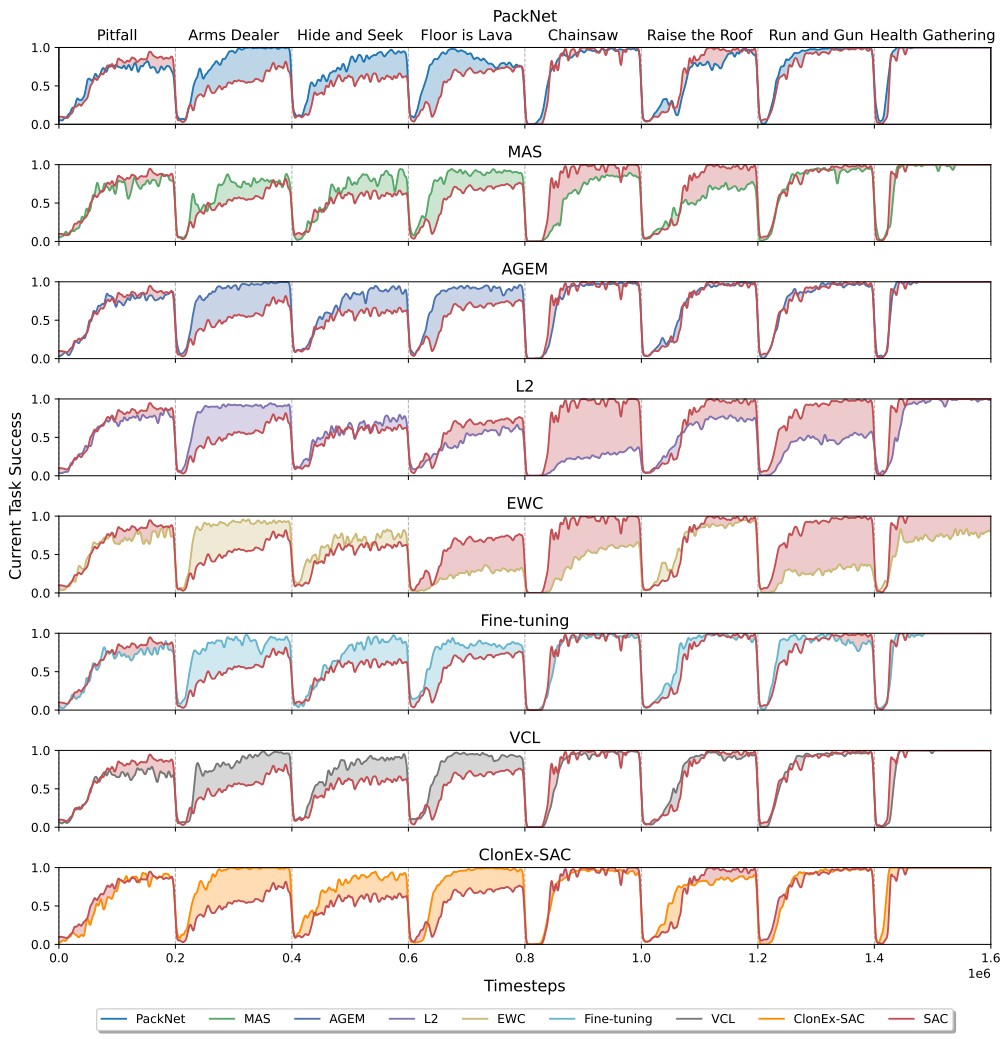

Figure 24: Forward transfer results on the CO8 sequence.

