# OpenReview forum: "COOM: A Game Benchmark for Continual Reinforcement Learning"
_NeurIPS.cc/2023/Track/Datasets_and_Benchmarks — NeurIPS 2023 Datasets and Benchmarks Poster_

### Official Review · Reviewer_eYzv · 2023-07-19
**Review for paper 572**

**Rating:** 6
**Confidence:** 4
**Correctness:** Yes
**Clarity:** The paper is well written

**Strengths:**

The paper presents the first benchmark dedicated to evaluating continual reinforcement learning (CRL) in complex 3D environments with diverse objectives and visuals. The authors created six unique VizDoom scenarios and applied multiple well-established continual learning (CL) methods to establish baseline performance on these tasks, consisting of 54 sequences. Through these experiments, the study offers a comprehensive examination of CL methods, assessing their effectiveness based on prominent CL criteria.

**Additional Feedback:**

The author can reference the aforementioned paper to gain deeper insights into Embodied AI simulators, which may serve as inspiration for future work involving these types of simulators.

1.Duan, Jiafei, Samson Yu, Hui Li Tan, Hongyuan Zhu, and Cheston Tan. "A survey of embodied ai: From simulators to research tasks." IEEE Transactions on Emerging Topics in Computational Intelligence 6, no. 2 (2022): 230-244.
2.Deitke, Matt, Dhruv Batra, Yonatan Bisk, Tommaso Campari, Angel X. Chang, Devendra Singh Chaplot, Changan Chen et al. "Retrospectives on the embodied ai workshop." arXiv preprint arXiv:2210.06849 (2022).

**Documentation:**

Yes

**Limitations:**

The author should further expand on the limitations instead of just a few words.

**Opportunities For Improvement:**

1. Based on Table 2, COOM and CRLMaze appear to have a similar structure, prompting the question of why CRLMaze couldn't be scaled up to achieve distinct differences between the two works.

2. While the author mentioned the use of embodied AI simulators in the related work, they did not provide sufficient justification for choosing VizDoom over other 3D environments with rich object and robot assets, such as AI2Thor or iGibson.

3. A key objective of introducing a new benchmark is to encourage further advancements in the field, but if existing CL methods already achieve a high performance, reaching 92% according to Table 3, it raises the question of how the benchmark can be further improved. Are there any insights into why certain tasks demonstrate significantly higher performance with existing CL methods?

If the author can address my concerns, i would consider to raise my rating.

**Relation To Prior Work:**

Yes, the paper did discuss about the prior work, however further discussion on why embodied AI simulators are not suitable.

**Summary And Contributions:**

The paper introduces COOM, a continual learning benchmark for embodied pixel-based RL. COOM is a 3D simulator designed to evaluate important aspects of continual RL, including catastrophic forgetting, forward knowledge transfer, and sample-efficient learning. The study also compares existing methods in continual learning using tasks derived from their simulator, which is built upon VizDoom. The primary goal of this benchmark is to provide the research community with a valuable and cost-effective challenge, facilitating a better understanding of the capabilities of current and future methods.

---

> ### Author Response · Authors · 2023-08-13
> **Addressing the Opportunities for Improvement Pt. 1**
>
> We genuinely appreciate the thorough review and the insightful observations made by the reviewer. We are gratified to hear that the paper's contributions and the COOM benchmark have resonated with you. Your feedback provides valuable insights and avenues for improvement. In our rebuttal, we will 1) clarify the core differences between CRLMaze and COOM, 2) explain why COOM is preferable over Unity3D based AI simulators, and 3) address the problem of lacking complexity in the benchmark.
>
> ### CRLMaze
> Indeed, CRLMaze[1] exhibits a noticeable similarity to the Cross-Domain (CD) setting within the COOM framework. The distinguishing factor, however, lies in the nature of the tasks. While CRLMaze tasks exhibit incremental difficulty, the environments in the CD sequence of COOM are not intentionally designed to grow in complexity across iterations. At best, CRLMaze could function as either:
> 1. A single scenario within the Cross-Objective sequences of COOM, featuring the object-picking task as its core. It is important to note that this competency is already embedded within certain scenarios (Health Gathering, Hide and Seek, Arms Dealer) in the current version of COOM. Consequently, introducing would not contribute much novelty.
> 2. The CD sequence. However, this adaptation would necessitate modifying the CRLMaze tasks of increased difficulty (M2, M3), and designing multiple additional environments.
>
> Instead, we chose to shape our environments based on the design principles of LevDoom[2], a generalization benchmark. We directly integrated certain LevDoom environments with visual modifications and drew inspiration from multiple scenarios, a facet we discussed in Section 4.2. Although CRLMaze is inherently tailored for Continual Learning, we found that the LevDoom generalization benchmark provided a more suitable foundation in terms of extensible environments for the development of a Continual Learning benchmark. We thus conclude to have obtained no discernible advantages in extending CRLMaze instead to further enrich the final outcome of our work.
>
> ### Embodied AI Simulators
> Continual RL is even more resource consuming than regular RL. Methods that exhibit a limited capacity for forward transfer need to adapt the policy almost from scratch to reach adequate performance on a newly presented task. This effectively multiplies the time and resources required by the number of tasks in the CL sequence, in contrast to solving a single RL problem. Hence, we have opted to optimize for a very lightweight and easily expandable simulation platform. Whereas ViZDoom is disentangled from real-life problems and lacks the visual fidelity and complexity of modern game engines like Unity3D (on top of which iGibson and AI2Thor are built), its simplicity and optimized design allow the simulation to run much faster. We believe this to be crucial for facilitating research on low-budget computational setups. Below we further discuss the aspect of lacking complexity.
>
> [1] V. Lomonaco et al. “Continual Reinforcement Learning in 3D Non-stationary Environments”, 2020
> [2] T. Tomilin et al. “LevDoom: A Benchmark for Generalization on Level Difficulty in Reinforcement Learning”, 2022
>
> The rebuttal is continued in the following comment.

---

> > ### Author Response · Authors · 2023-08-13
> > **Addressing the Opportunities for Improvement Pt. 2**
> >
> > ### Benchmark Complexity
> > We agree with the reviewer's concern that the benchmark seems to be solved (given our experimental results of some baselines on CO8 and CD8), and therefore does not encourage further advancements in the field. We have proposed three options to tackle this issue.
> > #### 1. Cross-Objective Challenge
> > We introduced the COC sequence as a significantly more complex version of CO8. Although it can be argued that solving this sequence presents more of an RL than a CL challenge, we posit that tasks of increased difficulty, which are inherently tougher to master, also introduce a greater level of complexity in terms of sustaining performance and achieving effective transfer. Therefore, we view the COC sequence as a valuable testing ground for future methods, as it effectively showcases both learning challenges and transfer difficulties.
> > #### 2. Performance Upper Bound
> > To facilitate a meaningful performance comparison across tasks within the Cross-Objective sequences, we have introduced a success metric that necessitates establishing an upper performance threshold for each scenario. If these bounds were proportionally raised for all scenarios, the complexity of sequences would also increase. We do not, however, see this as the best option to create a more complex challenge, as the approach is very artificial.
> > #### 3. Repeated Sequences
> > With the suggestion of another reviewer we have additionally included longer task sequences (CO16, CD16) in the benchmark by repeating their respective 8-task counterparts. We evaluated PackNet on CO16 and compare the results with CO8 in the following table.
> >
> > | Experiment   | Performance | Forgetting  | Transfer    |
> > |--------------|-------------|-------------|-------------|
> > | PackNet CO8  | 0.82 ± 0.09 | 0.03 ± 0.03 | 0.25 ± 0.05 |
> > | PackNet CO16 | 0.69 ± 0.20 | 0.05 ± 0.01 | 0.06 ± 0.30 |
> >
> > The challenge of CO16 posed by the repetition of the sequence becomes notably more demanding, as evidenced by the overall decline in metrics across all categories when compared to CO8. As algorithms advance in terms of sample efficiency and as computational resources become more abundant, we envision the potential to harness repeated sequences to elevate the benchmark's challenge and scalability. This forward-thinking approach sets the stage for future expansions and improvements that align with the evolving landscape of the field.
> >
> > #### Task complexity
> > When evaluating task complexity within the realm of continual learning, we posit that the focus should lie in identifying tasks that diverge the most from the rest in the sequence, instead of merely gauging the inherent complexity of individual tasks. Addressing this perspective, we delve into a brief exploration in Appendix D, wherein we assess the varying degrees of maintening performance across our scenarios of the Cross-Objective (CO) sequences. Our analysis of task-specific forgetting patterns reveals discernible distinctions in terms of performance retention; however, the disparities are not significant.
> >
> > We hypothesize that certain scenarios within the CO sequences exhibit heightened potential for sustaining performance due to their overlap in fundamental competencies (e.g., efficient navigation, eliminating enemies, collecting items). On the one hand, this grants a higher capicity for transfering acquired knowledge from previous tasks requiring similar competencies. On the other hand, decreases the likelihood of forgetting if the future tasks re-using these competencies. It is important to note that within the scope of our work, we have not run experiments to either validate or refute this assertion.
> >
> > ### Limitations
> > We believe the limitations of our work are quite extensively discussed in Appendix J. Perhaps the reviewer missed this, since there was no proper reference to this from the main paper. We have thus included a reference to the limitations in Section 4. If the reviewer finds that there are some shortcomings of our work that we have not sufficiently addressed, we are happy to include this.
> >
> > If we have managed to sufficiently address some or all of the shortcomings that the reviewer has mentioned, we would kindly ask the review to consider raising the score.

---

> > > ### Comment · Reviewer_eYzv · 2023-08-18
> > > **Thank you**
> > >
> > > I would like to thank the author for the rebuttal, i would take all these into consideration.

---

### Official Review · Reviewer_5vZ2 · 2023-07-20
**A useful benchmark with an adequate evaluation**

**Rating:** 7
**Confidence:** 4
**Clarity:** The paper is clearly written and easy…

**Strengths:**

1. This is the first benchmark for continual pixel-based RL that I am aware of.
2. Various approaches are evaluated, somewhat demonstrating the feasibility of handling some of the settings in the benchmark

**Additional Feedback:**

The following points are provided as feedback to hopefully help better shape the submitted manuscript, but did not impact my recommendation in a major way.

Abstract
- I wonder right away what the key insight of this benchmark is. Is it just another benchmark or is there something unique about it? I think it might be the first pixel-based continual RL benchmark, eg

Intro
- "First to benchmark image-based CRL in 3D environments". Maybe bring this up in the abstract?

Sec 3
- RL tasks are described as a POMDP but then the manuscript refers to images as states, where they should be observations
- This Section should mention that the task indicator is known by the RL agent.

Sec 4
- Overall, the benchmark description is clear and comprehensive. I get a sense that I know very well how things are implemented

Sec 4.2
- I like that they measure fw transfer as AUC. Missing reference to Continual World for defining this metric.

Layout
- Table 2 and table 1 should be flipped

Typos
- Line 1: reinforcement learning -> reinforcement learning (RL)
- Line 271: which is not particularly [in]efficient?

**Correctness:**

Apart from the points raised in the "Opportunities" section, the paper appears to be correct.

**Documentation:**

All the design choices of the benchmark are well documented.

**Ethics:**

No.

**Limitations:**

Limitations are only discussed in the appendix, and there is no mention of them (or reference to the appendix) in the main paper. The authors discuss (but do not address) the limitations mentioned in Opportunity 1.

**Opportunities For Improvement:**

1. In many ways, the design choices behind the benchmark and the evaluation mimic the design choices behind Continual World, inheriting some of the weaknesses of Continual World
    - All the baselines considered for evaluation are supervised methods adapted to the RL setting. A proper evaluation should consider continual RL methods specifically, which have been designed to handle the nuances of applying continual training in RL settings.
    - The task sequences are very short. It is certainly understandable that running over very long task sequences is impractical for expensive RL environments, but continual training over short sequences is not necessarily indicative of continual training over long sequences. How long (in wall-clock time) do the CD/CO8 evaluations take? Would it not be feasible to have a version with 20 or so tasks? Authors of future work could use short sequences for "development", and then run evaluations on the longer sequences
    - Unless there is a specific reason for considering a curriculum (e.g., tasks of increasing difficulty, or tasks containing shared skills somehow), continual learning methods should be evaluated over various task orderings, not just a fixed one. The reasoning is that using a random task order introduces randomness in approaches' performance, and by making it a _fixed_ random order all results will be affected by this randomness. This has been known since at least [1]
2. The single-head ablation provides very little insight. If two tasks solve for conflicting objectives (as described by the authors themselves), there is no way for a single network to make appropriate decisions for both tasks. This is well known and has been documented (e.g., [2]).
3. Given the similarities, the authors should more clearly credit the creators of Continual World for many of the design choices (not necessarily repeatedly, but at least acknowledge this somewhere).

[1] Ruvolo & Eaton. "ELLA: An Efficient Lifelong Learning Algorithm", 2013.

[2] Kessler et al. "Same State, Different Task: Continual Reinforcement Learning without Interference", 2022

**Relation To Prior Work:**

The related work (Section 2) discussion is comprehensive and fair.

**Summary And Contributions:**

The authors propose 5 task sequences to evaluate continual RL methods on 3D pixel-based embodied domains based on DOOM. The authors evaluate a set of baselines on the benchmark and discuss their findings.

---

> ### Author Response · Authors · 2023-08-21
> **Addressing the opportunities for improvement Pt. 1**
>
> Thank you for your comprehensive and insightful review. Your feedback is greatly appreciated, and we have carefully considered each of your suggestions to improve the quality and impact of our work. In our rebuttal, we provide detailed responses to your comments and outline the specific changes and additions we have implemented based on your recommendations.
>
> ### RL Dedicated Baselines
> To enhance the diversity of our baseline approaches, we have incorporated the ClonEx-SAC[1] method into our evaluation framework. This technique places a direct emphasis on refining CL through two key RL strategies: 1) applying behavioural cloning to optimize the actor policy of SAC, and 2) selecting one of the previously trained actor output heads (which yields the highest cumulative reward on the new task) for exploring a new task. We conducted a short hyperparameter search to ensure the best performance of the methods on our benchmark. The selected values are depicted in **bold**: (i) actor regularization coefficient [1e1, **1e2**, 1e3, 1e4, 1e5, 1e6, 1e7], (ii) gradient clipping [**None**, 0.1, 1.0], (iii) episodic memory batch size [64, **128**, 256]. The results on all sequences are presented in the following table. The values in **bold** denote instances where ClonEx-SAC outperforms all other baselines.
>
> | Sequence |   Performance   |   Forgetting    |    Transfer     |
> |----------|-----------------|-----------------|-----------------|
> | CD4      |   0.87 ± 0.03   |   0.00 ± 0.00   |   0.11 ± 0.05   |
> | CO4      |   0.86 ± 0.04   | **0.00 ± 0.02** |  -0.26 ± 0.09   |
> | CD8      | **0.92 ± 0.03** | **0.00 ± 0.02** |   0.13 ± 0.05   |
> | CO8      | **0.89 ± 0.07** |   0.01 ± 0.02   | **0.27 ± 0.10** |
> | COC      |   0.13 ± 0.05   |   0.02 ± 0.01   |   0.03 ± 0.01   |
> | Average  |   0.73          | **0.01**        |   0.06          |
>
> ClonEx-SAC emerges as the top-performning baselines on the long task sequences, and closely trails the best performing ones on the shorter sequences. Across all sequences, it notably ranks lowest in forgetting. For a visual comparison of ClonEx-SAC with other baselines, we include a plot depcting the Average Performance on sequences CO8 and CD8, accessible through this [link](https://www.imagebam.com/view/MENU0GQ).
>
> ### Long Task Sequences
> Our experiments on the CD8 and CO8 sequences average around 37 hours. However, we agree that it is useful to have longer sequences included in the benchmark to more adequately determine the long term continual learning capabilities. To this end, we further follow the design principles of Continual World, and in a similar fashion include longer sequences of 16 tasks (CD16, CO16) by repeating the respective shorter 8-task sequence. Although the tasks are repeated, this is not evident to the agent, as the task id’s are all unique, meaning that different policy output heads will be used in our default multi-head architecture design. Due to limited time for additional experiments, we only present the results of our best performing baseline (PackNet) on CO16.
>
> | Experiment   | Performance | Forgetting  | Transfer    |
> |--------------|-------------|-------------|-------------|
> | PackNet CO8  | 0.82 ± 0.09 | 0.03 ± 0.03 | 0.25 ± 0.05 |
> | PackNet CO16 | 0.69 ± 0.20 | 0.05 ± 0.01 | 0.06 ± 0.30 |
>
> The challenge of repeating a sequence becomes notably more demanding, as evidenced by the poorer metrics across all categories compared to a single iteration.  While PackNet continues to excel in counteracting forgetting, it does so at the cost of achieving lower performance in subsequent tasks. Additionally, the confidence intervals for performance and transfer are considerably broader compared to CO8, indicating heightened instability across multiple runs.
>
> Although we continue to consider the original 8-task sequences to remain at the core of our benchmark due to their adept balance between substantial continual learning challenges and manageable computational demands, it's worth noting that this brief experiment underscores the heightened difficulty posed by repeating a sequence. As algorithms progress in sample efficiency and computational resources become more abundant, we envision the possibility of leveraging repeated sequences to augment the challenge and scale of the benchmark. This forward-looking perspective paves the way for future extensions and enhancements as the field evolves.
>
> [1] M. Wołczyk et al. “Disentangling Transfer in Continual Reinforcement Learning”, 2022
>
> The rebuttal is continued in the following comments.

---

> ### Author Response · Authors · 2023-08-21
> **Addressing the opportunities for improvement Pt. 2**
>
> ### Task Ordering
> We completely agree with the reasoning that fixing the sequence of tasks may introduce a bias in the assessment of different approaches. However, randomising the order would only provide more equal grounds for comparison if the sequence is repeatedly executed across numerous trials, making the performance average out in the long run. Particularly in the context of RL, this swiftly becomes infeasible due to the already hefty walltime of a single sequence execution, as stated above. Considering the orientation of our work towards facilitating research within a moderately-budgeted computational setup, we have opted to adhere to fixed sequences. On the upside, all the methods have to tackle identical challenges of transfer from one fixed task to the following one in the sequence. While we acknowledge that the chosen task order lacks a specific rationale, we posit that this compromise offers a better balance between the trade-off of feasibility and minor potential imbalance.
>
> ### Single-Head Ablation
> It's important to note that despite conflicting objectives between two tasks in the Cross-Objective setup, substantial dissimilarity exists in their respective state spaces. This distinction results in the agent encountering a diverse range of observations from the environments. This diversity can form the foundation for the agent's ability to assimilate specific observable features even with a single network and output head, linking them to the value of a given state and the optimal action to be taken. We thus believe it is nevertheless valuable to determine which methods manage to function the best in such an ablated setting. Our experiments revealed that although regularisation-based methods indeed encountered great challenges in this context, PackNet achieved remarkably similar performance even when employing only a single output head. We would hence not fully rule out the capability of CL methods to function in this setting. The paper [2] also states that multi-head networks are certainly more effective in allowing to learn task specific parameters, but single-head networks are still commonly used as a more difficult baseline for CL benchmarks.
>
> ### Continual World
> To better acknowledge having adopted evaluation metrics and selection of CL methods from [3] and grant them more credit, we mention their work in our 3rd contribution: “3) Following the design principles of [3], we employ multiple well-known CL methods for baseline evaluations on our task sequences, assessing prominent CL criteria.”
>
> ### Minor Improvements/Corrections
> 1. Corrected section 3 (Preliminaries) to adequately formulate solving a POMDP via learning from pixel observations.
> 2. Swapped the order of Tables 1&2, and corrected the noted typos.
> 3. Referred to the benchmark limitations in the Appendix from the main paper.
> 4. Outlined the uniqueness and core findings of the paper in the abstract.
> 5. Section 5.1 now clarifies the agent’s awareness of tasks with: “For all of our baselines, we grant the agent full access to the task identity both at training and test time, a CL setting known as Task-Incremental Learning [4]”.
>
> [2] S. Kessler et al. “Same State, Different Task: Continual Reinforcement Learning without Interference.”, 2022
> [3] M. Wołczyk et al. “Continual world: A robotic benchmark for continual reinforcement learning”, 2021
> [4] L. Wang et al. “A Comprehensive Survey of Continual Learning: Theory, Method and Application”, 2023

---

> > ### Comment · Reviewer_5vZ2 · 2023-08-27
> >
> > Thank you for your responses.
> > - **RL baseline:** This is especially useful and I am glad that the authors were able to add this benchmarking experiment. I think it is a nice addition to the paper. Please be sure to include it in the main paper in the final version.
> > - **Long task sequences:** I am not sure I follow the motivation for sequencing together the same tasks. It is an artificial setting, and while it does allow us to study whether agents are capable of detecting repetitions, that point seems to be disconnected from the rest of the findings in this work. It may have been better to glue together two different 8-length sequences (e.g., CD8+CO8).
> > - **Task ordering:** The ordering could be varied for each random seed, instead of running various random seeds for each ordering. This is a common practice and allows us to simultaneously assess the effect of various sources of randomness. The imbalance may very well not be minor---often, the variance in the performance of one approach across different task sequences is very large, since most continual methods' performance depends heavily on the feature representation learned on the first task. As one additional note: the fact that the authors were not able to run experiments on multiple random orderings does not imply that the benchmark itself should be restricted to a single ordering. Maybe the authors could consider proposing that the _benchmark_ contain various random orderings, while clarifying that the _benchmarking experiments_ in the paper are limited to one single ordering.
> > - **Single-head ablation:** Thank you for the clarification. This is indeed a good point.
> >
> > On the whole, I think that the paper is stronger with these modifications and will increase my score accordingly. I do strongly encourage the authors to include a discussion of the limitation on task ordering in their final draft.

---

### Official Review · Reviewer_TUS9 · 2023-07-21
**COOM: A Game Benchmark for Continual Reinforcement Learning**

**Rating:** 7
**Confidence:** 4
**Correctness:** The content appears to be accurate.

**Strengths:**

This paper addresses the lack of comprehensive benchmarks in the subfield of continuous reinforcement learning, making a valuable contribution to ongoing research in this area.
The inclusion of various scenarios, visual differences, and difficulty levels within a single benchmark enables a wide range of experiments.
The paper thoroughly investigates and evaluates different types of continuous learning baselines.

**Additional Feedback:**

In Figure 1, CD8 and CO8 seem to be interchanged.

**Clarity:**

The description of the benchmark is well-written and provides sufficient understanding.

**Documentation:**

The detailed descriptions of each environment are well-explained. The code can be accessed via the provided link for verification.

**Ethics:**

No ethical concerns were identified.

**Limitations:**

Similar to the opportunities for improvement mentioned earlier, the paper lacks baselines specifically related to reinforcement learning.

**Opportunities For Improvement:**

Recent research has emphasized the importance of plasticity, the ability to adapt to new targets, in reinforcement learning [1,2,3,4]. Including these baselines would enhance the analysis and provide a comprehensive perspective.
[1] Abbas et al., "Loss of plasticity in continual deep reinforcement learning."
[2] Lyle et al., "Understanding plasticity in neural networks."
[3] Nikishin et al., "Deep Reinforcement Learning with Plasticity Injection."
[4] Lee et al., "Enhancing Generalization and Plasticity for Sample Efficient Reinforcement Learning."

**Relation To Prior Work:**

The disparities between this benchmark and previous ones are adequately delineated.

**Summary And Contributions:**

The authors introduce a novel benchmark called COOM, which focuses on continual reinforcement learning.
This benchmark consists of eight scenarios with different visual inputs, specific objectives, and two difficulty levels.
The authors propose three different settings of COOM: CO8, which presents sequentially different scenarios; CD8, which features a visually distinct environment in the same scenario; and COC, which incorporates varying levels of difficulty within the same scenario.
The paper defines an evaluation protocol and essential metrics, such as average performance, forgetting, and forward transfer, for continuous learning.
The authors provide results for different continuous learning methods.

---

> ### Author Response · Authors · 2023-08-21
> **Adressing Network Plasticity**
>
> ### Network Plasticity
> We thank the reviewer for suggesting to look into how our benchmark relates to network plasticity. This is indeed an interesting domain we have overlooked. Similar to the experimental design in [1], we aim to investigate whether our base SAC agent experiences a decline in network plasticity when confronted with the Cross-Objective task setting within our benchmark (the CO sequence exhibits a greater diversity across its tasks). To examine this, we utilize the CO4 and CO8 task sequences, cyclically exposing the agent to the sequence of tasks over 10 repetitions. We halve the the number of time steps of a single task, reducing them to 100K, as we found that this number is sufficient for convergence. The entire sequences thus lasts 4M and 8M environment iterations. After a task is finished, we maintain the network weights and only reset the learning rate decay and replay buffer. Insted of measuring Success (the unified metric), we assess the individual performance metric of each task. We wish to observe whether the initial performance achieved by the agent on a given task deteriorates over successive trials when re-encountering it.
>
> The figure depicting the learning curves of our results are accessible through the following [link](https://www.imagebam.com/view/MENJTMK). Intriguingly, unlike the trend observed with the Rainbow agent in [1], our findings diverge. On the shorter task sequences, the attained peak performance remains stable throughout the iterations. On CO8, there is a discernible dip in performance across most tasks midway through the entire sequence, yet notably, this decline is followed by a resurgence towards the end. We therefore conclude that our base SAC agent does not exhibit a decline in network plasticity when subjected to our benchmark, and more closely aligns with the *reset agent* coined in [1]. Consequently, we posit that, in the context of our benchmark, factors like plasticity, while important, might not be the primary concerns. Our findings strongly suggest that addressing challenges related to forgetting and transfer hold greater significance. The outcomes of our experiments underscore the difficulties encountered by several methods in these specific areas. Given this perspective, the inclusion of baselines specifically designed for sustaining network plasticity might not be of paramount importance.
>
> Moreover, some of the proposed baseline methods aimed at addressing plasticity are not directly applicable to our specific problem setting. For instance, the approach presented in [3] introduces a technique involving the introduction of a new set of randomly sampled parameters, and maintaining two copies of them. These copies are employed to learn a residual to the old network outputs, aiding in stability by using a bias term for predictions. However, it's important to note that this method is primarily designed for regular RL on a single task, rather than being tailored for the challenges posed by CL scenarios. While it aims to prevent performance plateaus, adapting this method to our CL context is not evident.
>
> [1] Z. Abbas et al. "Loss of plasticity in continual deep reinforcement learning", 2023
> [2] Nikishin et al. "Deep Reinforcement Learning with Plasticity Injection", 2023
>
> The rebuttal is continued in the following comment.

---

> ### Author Response · Authors · 2023-08-21
> **Addressing the lack of RL dedicated baselines**
>
> ### RL Dedicated Baselines
> To enhance the diversity of our baseline approaches, we have incorporated the ClonEx-SAC[1] method into our evaluation framework. This technique places a direct emphasis on refining CL through two key RL strategies: 1) applying behavioural cloning to optimize the actor policy of SAC, and 2) selecting one of the previously trained actor output heads (which yields the highest cumulative reward on the new task) for exploring a new task. We conducted a short hyperparameter search to ensure the best performance of the methods on our benchmark. The selected values are depicted in **bold**: (i) actor regularization coefficient [1e1, **1e2**, 1e3, 1e4, 1e5, 1e6, 1e7], (ii) gradient clipping [**None**, 0.1, 1.0], (iii) episodic memory batch size [64, **128**, 256]. The results on all sequences are presented in the following table. The values in **bold** denote instances where ClonEx-SAC outperforms all other baselines.
>
> | Sequence |   Performance   |   Forgetting    |    Transfer     |
> |----------|-----------------|-----------------|-----------------|
> | CD4      |   0.87 ± 0.03   |   0.00 ± 0.00   |   0.11 ± 0.05   |
> | CO4      |   0.86 ± 0.04   | **0.00 ± 0.02** |  -0.26 ± 0.09   |
> | CD8      | **0.92 ± 0.03** | **0.00 ± 0.02** |   0.13 ± 0.05   |
> | CO8      | **0.89 ± 0.07** |   0.01 ± 0.02   | **0.27 ± 0.10** |
> | COC      |   0.13 ± 0.05   |   0.02 ± 0.01   |   0.03 ± 0.01   |
> | Average  |   0.73          | **0.01**        |   0.06          |
>
> ClonEx-SAC emerges as the top-performning baselines on the long task sequences, and closely trails the best performing ones on the shorter sequences. Across all sequences, it notably ranks lowest in forgetting. For a visual comparison of ClonEx-SAC with other baselines, we include a plot depcting the Average Performance on sequences CO8 and CD8, accessible through this [link](https://www.imagebam.com/view/MENU0GQ).
>
>
> ### Figure 1
> Thank you for pointing out our mistake of mislabelling the task sequences (CD8 <---> CO8) in Figure 1. The correction will be included in the camera-ready version of the paper.
>
> [1] M. Wołczyk et al. “Disentangling Transfer in Continual Reinforcement Learning”, 2022

---

### Official Review · Reviewer_u5Cf · 2023-07-21
**This paper is well written, the details are clearly explained, and the experiments are well organized.**

**Rating:** 7
**Confidence:** 3
**Clarity:** This paper and the readme document ar…

**Strengths:**

This paper is well written, the details are clearly explained, and the experiments are sufficient and well organized.

**Additional Feedback:**

n/a

**Correctness:**

The benchmark is well constructed and easily accessible from [GitHub](https://anonymous.4open.science/r/COOM-3927/README.md).

**Documentation:**

The supplemental document is sufficient, well writen and organized.

**Ethics:**

This paper provides a reinforcement learning benchmark in simulation with no ethical concerns.

**Limitations:**

Yes, the authors analysis the limitation of COOM in the Appendix.

**Opportunities For Improvement:**

Based on my evaluation, I recommend that you swap the positions of Table 1 and Table 2. The current first reference to Table 2 appears in the first paragraph on page 3, but Table 2 is not presented until page 4. By moving Table 2 ahead of Table 1, this issue can be resolved.

**Relation To Prior Work:**

Yes. The author analyzes the limitations of previous work and proposes DOOM as its solution.

**Summary And Contributions:**

This paper proposed COOM, a continual learning benchmark for embodied pixel-based RL. The author claimed that this is the first benchmark specifically targeted towards CRL in complex 3D environments with differing objectives and visuals. The mainly contributions are three-fold:
1. COOM assembles 4 base task sequences for further tightening the experimental loop. A very complex sequence is also provided as a challenge for advanced future algorithms.
2. COOM provides 6 novel ViZDoom scenarios with contrasting visuals and dynamics.
3. This paper employs multiple well-known CL methods for basedline evaluations on the benchmark.

---

> ### Author Response · Authors · 2023-08-13
>
> We thank the reviewer for the suggestion of swapping Tables 1 and 2. This indeed induces more coherency to the logical flow of the paper. We will include the change in the camera-ready version. In case there are no other flaws or shortcomings for us authors to address, we would kindly ask the reviewer to consider raising the score.

---

### Official Review · Reviewer_vkRe · 2023-07-21
**A well-written work that proposes a continual RL benchmark**

**Rating:** 6
**Confidence:** 4
**Correctness:** I think the claims are all correct.
**Clarity:** Yes, it is well written.

**Strengths:**

1. The writing is good and the narration is easy to follow and understand.
2. The experiments are comprehensive and provide a base for future continual RL study.
3. The proposed three task sequences are interesting, it is good to test how robust CRL methods are to different level of task interference.

**Additional Feedback:**

My suggestions for improvement are listed in the improvement section.

**Documentation:**

I do not see any documentation yet, though it is promised it will be well-documented. (please correct me if I am wrong)

**Limitations:**

Yes.

**Opportunities For Improvement:**

I believe this work provides yet another benchmark for studying continual RL. But it is less clear to me why people would use this particular benchmark instead of the existing one, like those on metaworlds, habitat or atari. I was expecting to see some surprising findings that could be particularly interesting to either researchers from the continual learning or reinforcement learning communities. How regularizing critics can be harmful to the CL algorithms is pretty interesting. I would suggest the authors do something similar, focusing more on the *RL* part. For instance, as the proposed benchmark is an image-based benchmark, does it matter what base encoder we use for continual learning? Do techniques like prioritized experience replay help? Will online algorithms perform better than offline algorithms? Addressing those questions could be much more interesting than simply saying that PackNet performs the best (which is certainly expected). I understand that for the benchmarking purpose, it is necessary to go through several continual learning methods. But as a benchmark, it should provide some new insights or new opportunities for the community.

**Relation To Prior Work:**

It discusses prior works carefully, but it is less clear how the proposed benchmark stands out in any direction.

**Summary And Contributions:**

The work introduces a novel benchmark based on VisDoom domain for studying continual RL. The authors apply standard continual learning algorithms on top of SAC and benchmark the results. The finding that critic regularization can be harmful in most cases is interesting. Other than this finding, most findings are expected in this work and are not specific to the RL domain.

---

> ### Author Response · Authors · 2023-08-13
> **Addressing the lack of insights Pt. 1**
>
> Thank you for the comprehensive review and valuable feedback on our work. We foremost acknowledge the need to differentiate our benchmark and justify its relevance within the broader landscape of existing benchmarks. We agree that valuable insights strengthen the benchmark's contribution to the research community. Based on your suggestions, we have further extended our experimental analysis to include 1) prioritizing experience, 2) a different base encoder and 3) data augmentation.
>
> ### Prioritized Experience Replay
> To investigate the impact of prioritizing experiences on our benchmark, we employed PER to some of our baselines and ran them on the CO4 task sequence. For all methods, we assign higher sampling probabilities to experiences with larger TD errors. For AGEM, we additionally use the weights assigned by PER when storing experiences of the previous task from the episodic memory buffer. We used the hyperparameters from [1] for the proportional variant.
>
>
> | Method  | Performance -PER | Forgetting -PER | Transfer -PER | Performance +PER | Forgetting +PER | Transfer +PER |
> |---------|----------------------------|-------------------------|--------------------|------------------------|----------------------|----------------|
> | PackNet | 0.87 ± 0.07                | 0.01 ± 0.00             | -0.24 ± 0.32       | 0.85 ± 0.05            | 0.03 ± 0.00          | -0.00 ± 0.06   |
> | MAS     | 0.72 ± 0.08                | 0.24 ± 0.00             | -0.04 ± 0.06       | 0.55 ± 0.13            | 0.52 ± 0.00          | -0.35 ± 0.15   |
> | AGEM    | 0.42 ± 0.13                | 0.80 ± 0.00             | 0.03 ± 0.10        | 0.36 ± 0.07            | 0.91 ± 0.04          | -0.05 ± 0.09   |
> | L2      | 0.80 ± 0.08                | 0.00 ± 0.00             | -0.60 ± 0.37       | 0.40 ± 0.26            | 0.00 ± 0.00          | -1.73 ± 0.83   |
>
>
> Surprisingly, all the methods we tested had worse performance, especially L2. PackNet was nearly unchanged, and even improved in terms of transfer. For better comparison we also include a table with the gain/loss compared to the original setting.
>
> | Method  | Performance         | Forgetting           | Transfer       |
> |---------|---------------------|----------------------|----------------|
> | PackNet |    -1.44%           |  +0.02               |  +0.24         |
> | MAS     |   -24.24%           |  +0.28               |  -0.31         |
> | AGEM    |   -15.03%           |  +0.10               |  -0.09         |
> | L2      |   -49.79%           |  -0.00               |  -1.13         |
>
> [1] T. Schaul et al. “Prioritized experience replay”, 2016
>
> The rebuttal is continued in the following comments.

---

> > ### Author Response · Authors · 2023-08-13
> > **Addressing the lack of insights Pt. 2**
> >
> > ### Visual Encoding
> > Although it is a very interesting question what effect base visual encoders have on continual learning, our reinforcement learning setting is not ideal for investigating the SOTA encoders from CV. Our RL environments involve low resolution images (160x120, which we further downscale to 84x84) and less complex visual patterns compared to real-life images from e.g. ImageNet, which can make using large-scale models (ResNet, VGG, ViT) unnecessary. Additionally, the increased model complexity further increases the computational costs and training times, creating a high opportunity cost for rather insignificant benefits.
> >
> > We do, however, consider a more simple architectural variation to our model. Since our environments have sequential dependencies in the observations, temporal information plays a crucial role in decision-making. We combine our CNN architecture from [1] with an RNN. More precisely, instead of sending stacked frames through the CNN, we now extract spatial features from individual observations and process the flattened features with a 512-unit LSTM layer before concatenating with the one-hot encoded task id vector. We wish to observe how this approach extends to the CL forgetting and transfer conundrum. We evlaute the same methods on CO4 as in the previous section and present the results in a similar fashion.
> >
> > | Method  | Performance CNN | Forgetting CNN | Transfer CNN | Performance CNN+LSTM | Forgetting CNN+LSTM | Transfer CNN+LSTM |
> > |---------|-------------------------|--------------------------|--------------------|----------------------|-----------------------|-----------------|
> > | PackNet |  0.87 ± 0.07            | 0.01 ± 0.00              | -0.24 ± 0.32       | 0.83 ± 0.06          | 0.03 ± 0.00           | -0.21 ± 0.07    |
> > | MAS     |  0.72 ± 0.08            | 0.24 ± 0.00              | -0.04 ± 0.06       | 0.01 ± 0.01          | 0.00 ± 0.00           | -3.05 ± 0.06    |
> > | AGEM    |  0.42 ± 0.13            | 0.80 ± 0.00              | 0.03 ± 0.10         | 0.48 ± 0.09          | 0.74 ± 0.00           | -0.27 ± 0.03    |
> > | L2      |  0.80 ± 0.08            | 0.00 ± 0.00              | -0.60 ± 0.37       | 0.01 ± 0.01          | 0.00 ± 0.00           | -3.07 ± 0.01    |
> >
> >
> > | Method  | Performance         | Forgetting            | Transfer        |
> > |---------|---------------------|-----------------------|-----------------|
> > | PackNet |    -4.52%           |  +0.02                |  +0.04          |
> > | MAS     |   -98.59%           |  -0.24                |  -3.02          |
> > | AGEM    |   +14.89%           |  -0.06                |  -0.30          |
> > | L2      |   -98.45%           |  -0.00                |  -2.47          |
> >
> > The outcomes for the regularization-based techniques were notably discouraging, as they were completely unable to reach any meaningful performance. This prompts us to hypothesize that weight regularization might not be suitable when applied to RNNs. Interestingly, PackNet encountered a marginal decline in performance, while AGEM displayed a substantial enhancement, albeit at the expense of decreased forward transfer. We hypothesize that with AGEM, the LSTM's ability to store and manage episodic memories enhances its capacity to prevent catastrophic forgetting when transitioning between tasks, resulting in improved overall performance compared to a single CNN architecture.
> >
> > The rebuttal is continued in the following comments.

---

> > > ### Author Response · Authors · 2023-08-13
> > > **Addressing the lack of insights Pt. 3**
> > >
> > > ### Image Augmentation
> > > We believe that image augmentation can offer a diverse range of benefits CRL. By enabling the model to perceive objects and patterns in various contexts, augmentation can enhance its adaptability. It can introduces variability into the input data, playing a crucial role in averting catastrophic forgetting. Moreover, augmentation can emulate the complexities posed by novel tasks, equipping the model to minimize the adverse effects of task interference. Finally, it can act as a robust regularizer, reducing the risk of overfitting to specific patterns and bolstering the model's capacity to manage task variations effectively. In our analysis, we explore three augmentation methods to harness these advantages.
> > >
> > > #### 1. Random Convolution
> > > Lee et al. [2] show that application of a random convolutional layer to observations during training improve generalization in 3D navigation tasks. We wish to observe whether this extends to our setting in continual learning
> > > #### 2. Random Shift
> > > We use the augmentation of the DrQ [3] method. This entails padding each side of the image by 4 pixels (by repeating boundary pixels) and then selecting a random 84 × 84 crop, yielding the original image shifted by ±4 pixels. The authors found the random shifts to reduce overfitting and enable SAC to thrive without the need for auxiliary losses.
> > > #### 3. Random Noise
> > > Tobin et al. [4] consider adding random Gaussian noise (mean=0, std=0.1) to the image as a method of data augmentation.
> > >
> > > #### Experiments
> > > We apply the listed image augmentation methods to three of our baselines from different CL families and evaluate them on the CO4 sequence.
> > >
> > > ##### Performance
> > >
> > > | Method  | No Aug               | Conv                 | Shift                | Noise                |
> > > |---------|----------------------|----------------------|----------------------|----------------------|
> > > | PackNet | 0.87 ± 0.07          | 0.71 ± 0.07          | 0.87 ± 0.04          | 0.88 ± 0.03          |
> > > | MAS     | 0.72 ± 0.08          | 0.08 ± 0.07         | 0.51 ± 0.09         | 0.56 ± 0.09         |
> > > | VCL     | 0.40 ± 0.14          | 0.05 ± 0.04         | 0.34 ± 0.04         | 0.38 ± 0.05         |
> > >
> > >
> > > ##### Forgetting
> > >
> > > | Method  | No Aug               | Conv                 | Shift                | Noise                |
> > > |---------|----------------------|----------------------|----------------------|----------------------|
> > > | PackNet | 0.01 ± 0.00          | 0.00 ± 0.02          | 0.00 ± 0.01          | 0.00 ± 0.00          |
> > > | MAS     | 0.24 ± 0.00        | 0.00 ± 0.00         | 0.62 ± 0.03         | 0.58 ± 0.02         |
> > > | VCL     | 0.82 ± 0.00        | 0.01 ± 0.01         | 0.79 ± 0.00         | 0.91 ± 0.00         |
> > >
> > > ##### Transfer
> > >
> > > | Method  | No Aug               | Conv                 | Shift                | Noise                |
> > > |---------|----------------------|----------------------|----------------------|----------------------|
> > > | PackNet | -0.24 ± 0.32         | -1.05 ± 0.15         | -0.11 ± 0.02         | -0.13 ± 0.02         |
> > > | MAS     | -0.04 ± 0.06        | -2.68 ± 0.30        | -0.22 ± 0.05        | -0.13 ± 0.02        |
> > > | VCL     | -0.57 ± 0.30        | -2.81 ± 0.17        | -0.25 ± 0.12        | -0.12 ± 0.03        |
> > >
> > > Unfortunately, our experimentation with these augmentations did not yield any noticeable improvements on our selected baselines. On the contrary, the effect was deemed adverse in most cases. Random convolutions proved to be excessively robust as an augmentation method, significantly disrupting the learning process of all methods. Conversely, the remaining techniques led to marginal improvements in PackNet's performance, while inducing a decline in performance for the other two methods.
> > >
> > > [1] T. Schaul et al. "Prioritized experience replay", 2016
> > > [2] K. Lee et al. "A simple randomization technique for generalization in deep reinforcement learning", 2019
> > > [3] I. Kostrikov et al. "Image Augmentation Is All You Need: Regularizing Deep Reinforcement Learning from Pixels", 2020
> > > [4] J. Tobin et al. "Domain Randomization for Transferring Deep Neural Networks from Simulation to the Real World", 2017
> > >
> > > The rebuttal is continued in the following comment.

---

> > > > ### Author Response · Authors · 2023-08-13
> > > > **Addressing the lack of insights Pt. 4**
> > > >
> > > > #### Offline RL / On-policy RL
> > > > We find the reviewer's suggestion regarding exploring offline RL methods somewhat perplexing. Our benchmark is distinctly tailored for online RL, wherein agents learn through direct interactions with the environment. Offline algorithms, however, hinge on pre-existing datasets, which we have not composed. It is conceivable that the reviewer meant a comparison between on-policy methods and our off-policy SAC implementation. However, it is important to note that on-policy methods, like PPO, are known to be sample-inefficient in environments with costly trajectory collection. This inefficiency results in significantly longer training times and places a high computational burden on the learning process. Furthermore, some of our continual learning baselines necessitate a replay buffer, which is a fundamental component in off-policy algorithms. As on-policy methods lack a replay buffer, their integration into our benchmark would require substantial modifications and might not align with the primary focus of our work.
> > > >
> > > > #### Documentation
> > > > The statement in the paper regarding the project being well-documented implies that our GitHub repository is equipped with a comprehensive README that provides coherent instructions for installation and usage of the benchmark. Moreover, we have made a conscious effort to ensure that the code is thoroughly documented, making it easy to understand and navigate.

---

### Decision · Program_Chairs · 2023-09-22

**Decision:**

Accept (Poster)

**Comment:**

This paper introduces a new benchmark for continual learning (CL) in RL based on ViZDoom along with evaluation of several baseline CL methods. Reviewers raised several good points during the discussion -- evaluating plasticity, inclusion of RL baselines, task sequence ordering and length, whether the benchmark is close to 'solved' already -- which the authors had reasonable responses to and the paper is indeed stronger / more complete as a result. I'd encourage the authors to incorporate these updates to the main paper. I also appreciate the thorough discussion on limitations in Appendix J. Overall, this is a solid contribution. I recommend acceptance.